

# Technical note: NASAaccess – A tool for access, reformatting, and visualization of remotely sensed earth observation and climate data

Ibrahim Nourein Mohammed[1,2,3], Elkin Giovanni Romero Bustamante[4], John Bolten[1], E. James Nelson[4]

[1]Hydrological Sciences Laboratory, NASA Goddard Space Flight Center, Mail Code 617.0, Greenbelt, MD 20771, USA
[2]Science Applications International Corporation, 12010 Sunset Hills Road, Reston, VA, 20190, USA
[3]Environmental Sciences and Policy Program, Johns Hopkins University, 1717 Massachusetts Ave NW, Washington, D.C., 20036, USA
[4]Civil and Construction Engineering Department, Brigham Young University, Provo, UT, 84602, USA

*Correspondence to*: Ibrahim N. Mohammed (ibrahim.mohammed@nasa.gov)

**Abstract.** NASA has launched a new initiative; the Open-Source Science Initiative (OSSI) to enable and support science towards openness. The OSSI initiative supports open-source software development and dissemination. In this work, we present *NASAaccess* which is an open-source software package and web-based environmental modeling application for earth observation data accessing, reformatting, and presenting quantitative data products. The main objective of developing the *NASAaccess* platform is to facilitate exploration, modeling, and understanding of earth data for scientists, stakeholders, and concerned citizens whose objectives align with the new OSSI goals. The *NASAaccess* platform is available as software packages (i.e., R and conda packages) as well as an interactive format web-based environmental modeling application for earth observation data developed with the Tethys Platform. The *NASAaccess* has been envisioned to lower the technical barriers and simplify the process of accessing scalable distributed computing resources and leverage additional software for data and 20   computationally intensive modeling frameworks. Specifically, *NASAaccess* is developed to meet the need for seamless earth observation remote sensing and climate data ingestion into various hydrological modeling frameworks. Moreover, *NASAaccess* is also contributing to keep interested parties and stakeholders engaged with environmental modeling, accessing the information available in various remote sensing products. *NASAaccess* current capabilities covers various NASA datasets and products that include the Global Precipitation Measurement (GPM) data products, the Global Land Data Assimilation System 25   (GLDAS) land surface states and fluxes, and the NASA Earth Exchange Global Daily Downscaled Projections (NEX-GDDP) Coupled Model Intercomparison Project Phase 5 (CMIP5) & Coupled Model Intercomparison Project Phase 6 (CMIP6) climate change dataset products.



## 1 Introduction

One of the key elements of a paradigm shift in hydrologic science as outlined by Wagener et al., (2010) is real-time learning

observations, modeling, and management are interactive exercises with feedback and updating. Recently, sharing data, code, and other research products has become more common, but is still not a popular practice. That is because there are few incentives for preparing datasets and code for sharing and may even be discouraged by current programs and agencies who are hesitant to support data sharing platforms. This is one of the limitations to the progress of science as discussed by a recent National Academies of Sciences, Engineering, and Medicine report (National Academies of Sciences Engineering and

Medicine (U.S.), 2018).

Xu et al., (2022) presented an overview of visual computing applications developed for water resources management. These numerous applications have led to the emergence of innovative Big Data applications that can address past challenges and generate useful insights in water science disciplines (Talia et al., 2016). Xu et al., noted that many past visual computing applications developed for water resources management, integrated visual computing techniques into GIS, cyberinfrastructure,

and domain models to benefit the big data analysis aspect of water resources management. These new visual computing techniques and features are then becoming effective tools for disseminating water education, raising public awareness of various water problems, and increasing public engagement. For instance, the Enhancing National Climate Services (ENACTS) initiative led by Columbia University's International Research Institute for Climate and Society (IRI) has been making efforts to disseminate climate information and support developing countries decision makers, and stakeholders in making climate-

sensitive economic activities more resilient to current climate extremes and adapting to the changing climate (Nsengiyumva et al., 2021). ENACTS is an initiative developed to alleviate the challenges of climate data availability, as well as access and use by supporting countries to generate high-resolution gridded climate data time series and derived climate information products that are readily accessible to decision-makers (Dinku et al., 2014;Dinku et al., 2018).

Earth science data observations are archived at the National Aeronautics and Space Administration (NASA) Goddard Earth

Sciences Data and Information Services Center (GES DISC) and other NASA data centers. The data observations are primarily organized as time-step arrays and in several common formats that support the creation, access, and sharing of array-oriented scientific data (e.g., HDF, netCDF). Ongoing work has been done over the years to facilitate the access to, use of, and meet

the need of NASA data by providing tools and services for data visualization, sub-setting, and format conversion. In Table 1,

we summarize a few NASA GES DISC tools and services that have been developed to meet growing needs and applications

expressed by users for remote sensing earth observation data.

NASA has launched a new initiative; the Open-Source Science Initiative (OSSI) to enable and support science towards

openness (https://science.nasa.gov/open-science-overview). The OSSI supports open-source software development and

dissemination. To help meet challenges to the progress of science in earth observation data access and management more tools

need to be available, accessible, and understandable. This manuscript describes an open-source platform, e.g., *NASAaccess*

package, for accessing, and presenting quantitative remote sensing earth observation, and climate data products in an

interactive format so that scientists, stakeholders, and concerned citizens can engage in the exploration, modeling and

understanding of the data. The *NASAaccess* platform is available as R (R Development Core Team, 2022) and conda

(https://docs.conda.io/en/latest/) software packages as well as an interactive format web-based environmental modeling

application for earth observation data developed in the Tethys Platform framework (Swain et al., 2016). The *NASAaccess* is

envisaged to lower the technical barrier and simplify the process of accessing scalable distributed computing resources and

leverage a wide array of satellite-based earth observations for more comprehensive computationally intensive modeling

frameworks. Specifically, *NASAaccess* is developed to meet the need for seamless earth observation remote sensing and

climate data ingestion into other modeling frameworks including the Variable Infiltration Capacity - VIC (Liang et al., 1994),

the Distributed Hydrology Soil Vegetation Model - DHSVM (Wigmosta et al., 1994), the Regional Hydro-Ecologic Simulation

System (RHESSys) model (Tague and Band, 2004), and the Soil and Assessment Water Tool - SWAT (Arnold and Fohrer,

2005). Moreover, *NASAaccess* is also contributing to keep interested parties and stakeholders engaged with environmental

modeling, accessing the information available in various remote sensing products.

## 2 Methodology

### 2.1 NASAaccess Key functionalities

The current *NASAaccess* (v.3.3.0) capabilities covers various National Aeronautics and Space Administration (NASA) datasets

and products that include the Global Precipitation Measurement (GPM) data products (Huffman et al., 2019), the Global Land



Data Assimilation System (GLDAS) land surface states and fluxes (Rodell et al., 2004), and the NASA Earth Exchange Global

Daily Downscaled Projections (NEX-GDDP) Coupled Model Intercomparison Project Phase 5 (CMIP5) (Wood et al.,

2002;Wood et al., 2004;Maurer and Hidalgo, 2008;Thrasher et al., 2012) & Coupled Model Intercomparison Project Phase 6

(CMIP6) (Thrasher et al., 2022) climate change dataset products. A brief description is given for the current *NASAaccess*

(v.3.3.0) function capabilities in Figure 1. In principle, the functionality of the *NASAaccess* can be summarized as:

    a)  Accessing the NASA servers to download earth observation data by fetching specific data for a specific domain and

        period,

    b)  Clipping the needed data grids to an input shapefile of a user study watershed,

c)  Handling any temporal (i.e., processing diurnal minimum and maximum air temperatures from hourly input data) or

        spatial (e.g., finding the data that corresponds to the study area centroid) inconsistencies,

    d)  Generating gridded data files and definition files compatible with the various hydrological models (i.e., ascii format).

## 2.2 NASAaccess Package Requirements

The *NASAaccess* Package needs Earthdata Login credentials (https://urs.earthdata.nasa.gov/) to be operable. Earthdata is a

user registration and user profile management system for users getting Earth science data from any of the Distributed Active

Archive Centers (DAACs) that comprise NASA's Earth Observing System Data and Information System (EOSDIS). The

*NASAaccess* Package relies on the *curl* tool to transfer data from NASA servers to a user machine, using HTTPS supported

protocol. The *curl* package (https://github.com/jeroen/curl) provides bindings to the libcurl C library for R software program

(R Development Core Team, 2022). The *curl* package supports retrieving data in-memory, downloading to disk, or streaming

using the R "connection" interface. The *curl* command embedded in *NASAaccess* is designed to work seamlessly by appending

appropriate logon information to the ".netrc" file and the cookies file ".urs_cookies" to fetch various data products. The ".netrc"

and ".urs_cookies" files need to be stored at the user local directory before running any *NASAaccess* function, otherwise the

requested data will not be retrieved. Further details on how to make *curl* tool work with *NASAaccess* package and how to

create the ".netrc" file and the ".urs_cookies" cookies file can be reviewed at the *NASAaccess* Open Science Framework (OSF)

wiki pages https://osf.io/ctj2k/wiki/home/.



### 2.3 Tethys Application Framework

Tethys Platform (Swain et al., 2015;Swain et al., 2016) is a development and hosting environment for environmental web applications. The Tethys Platform consists of three major components: Tethys Software Suite, Tethys Software Development Kit (SDK), and Tethys Portal. An overview of the Tethys Platform and links to documentation, bug reporting, and support

forum are available online at http://www.tethysplatform.org. The Tethys platform has created a common medium for scholars and scientists enabling them to envision, develop, and deploy several notable Earth Observation web applications (McStraw et al.;McDonald et al., 2019;Nelson et al., 2019;Qiao et al., 2019;Saah et al., 2019;Gan et al., 2020;Bustamante et al., 2021;Khattar et al., 2021;Sanchez Lozano et al., 2021). The application structure for the *NASAaccess* Tethys web application uses the Model-View-Controller (MVC) software architecture discussed in McDonald et al., (2019). The Tethys platform uses

a PostgresQL database to store the data of each installed application. The model's module in a Tethys application is responsible for defining the different database tables structure, which later will be initialized by a custom script. In the case of the *NASAaccess* application, the Tethys platform creates and assigns a database to the *NASAaccess* application, but no tables are created because the *NASAacess* application does not define a data model. In other words, the data that the *NASAaccess* application fetches and retrieves is not saved in the PostgresQL associated with the application, rather it is just downloaded by

the user when it is ready. The controllers defined for the *NASAaccess* Tethys web application use the *NASAaccess* conda (https://docs.conda.io/en/latest/) package (`r-nasaaccess`) that handles the logic and functionality of the web application to connect and retrieve the specified data from the NASA servers. The controller module uses the `r-nasaaccess` through a conda installation instead of the Comprehensive R Archive Network (CRAN) https://CRAN.R-project.org or a Github installation of the `r-nasaaccess` because the Tethys platform works within a conda environment

(https://docs.conda.io/projects/conda/en/latest/user-guide/concepts/environments.html). As a result, using the `r-nasaaccess` conda package is compatible with the conda environment in which the *NASAaccess* application was installed. The use of the `r-nasaaccess` in the controller module is through the subprocess python library that calls an R-script to fetch the data and notify the user via email. The view modules represent the HTML pages that are rendered for the users to see and include necessary web-based GIS mapping functionalities. In the case of the *NASAaccess* application, the view module

allows the user to input and visualize shapefile and TIF files that will be used with the `r-nasaaccess` conda package. The

module view will also render plots associated with the data fetched by the `r-nasaaccess` conda package. The *NASAaccess* Tethys web application flow chart is depicted in Figure 2. From the left, we see that the current *NASAaccess* version (v.3.3.0) access different data products from NASA EARTHDATA portal (https://www.earthdata.nasa.gov/) such as GPM, GLDAS and different downscaled climate change data products. The controller's modules fetch data through the different methods in

the `r-nasaaccess` conda package in the conda-Forge channel (https://anaconda.org/conda-forge/r-nasaaccess). After reading the user study area shapefile and a digital elevation model raster for the study area, the *NASAaccess* Tethys application produces reformatted and clipped remotely sensed earth observation or climate change data products. Once the job is finished the *NASAaccess* Tethys application notifies the user with a reminder email with a unique code referring to the selected data requests. The *NASAaccess* Tethys application allows for data visualization and sharing. On that note, the *NASAaccess* Tethys

application facilitates data visualization and downloading for users who are interested so that further data analysis can be performed. On the far right, we see the *NASAaccess* Tethys Software Development Kit which includes a snapshot of the *NASAaccess* Tethys application home window with various data visualization examples to illustrate the utility of the application.

In summary, the *NASAaccess* Tethys application gives time series and spatial mapping visualization features for all the

functions available. Moreover, the user of the *NASAaccess* Tethys application receives the requested data formatted and ready to be ingested into other modeling frameworks such as the Variable Infiltration Capacity - VIC (Liang et al., 1994), the Distributed Hydrology Soil Vegetation Model - DHSVM (Wigmosta et al., 1994), the Regional Hydro-Ecologic Simulation System (RHESSys) model (Tague and Band, 2004), and the Soil and Assessment Water Tool - SWAT (Arnold and Fohrer, 2005) hydrological modeling frameworks. Another feature that *NASAaccess* Tethys application supports is the ability to

visualize and inspect different datasets processed by different functions at a specific watershed during one or different time periods in one job. This feature is useful when the user is interested in studying the impacts of climate change or any other hydrological regime changes.





### 2.4 NASAaccess Installation Steps

### 2.4.1 R Software

On a local machine the user should have installed the following programs as well as setting up a user account. The list below

gives a summary of what is needed to be done prior to working with *NASAaccess* software on any local machine.

1.   Installing R software (https://www.r-project.org/)

2.   Installing Rstudio software (https://posit.co/) (Optional)

3.   *NASAaccess* R package needs a user registration access with Earthdata (https://www.earthdata.nasa.gov/). Users

should set up a registration account(s) with Earthdata login as well as authorizing NASA GES DISC data access.

Please refer to https://disc.gsfc.nasa.gov/data-access for further details.

4.   After registration with Earthdata *NASAaccess* software package users should create a reference file ("*.netrc"*) with

Earthdata credentials stored in it to streamline the retrieval access to NASA servers.

o   Creating the "*.netrc"* file at the user machine *Home* directory and storing the user NASA GES DISC logging

information in it is needed to execute the *NASAaccess* package commands. Accessing data at NASA servers

is                           further                         explained                               at

https://wiki.earthdata.nasa.gov/display/EL/How+To+Access+Data+With+cURL+And+Wget

o   For Windows users the NASA GES DISC logging information should be saved in a file "_netrc" beside the

".netrc" file explained above.

5.   Installing *curl* software. Since Mac users have *curl* as part of macOS build, Windows users should make sure that

their local machines build have *curl* installed properly.

6.   Checking if you can run *curl* from your command prompt. Type `curl --help` and you should see the help pages

for the *curl* program once everything is defined correctly.

7.   Within Rstudio or R terminal run the following commands to install *NASAaccess*:

```
library(devtools)
install_github("nasa/NASAaccess", build_vignettes = TRUE)
library(NASAaccess)
```





### 2.4.2 Conda Environment

Like the R software, *NASAaccess* conda package (r-nasaaccess) needs a user registration access with Earthdata

(https://www.earthdata.nasa.gov/) and storing those credentials in the reference file ".netrc" as well as creating a cookies file

".urs_cookies". If the user has successfully prepared the needed steps to run the *NASAaccess* R package (i.e., creating

registration access and storing it in a local machine), then there is no need to duplicate these steps here again. Installing the r-

nasaaccess in a conda environment allows users to have packages in different programming languages due to the language

interoperability of the conda environment. To install the *NASAacces*s package in python (r-nasaaccess), run the following

syntax in a Python terminal:

```
conda install -c conda-forge r-nasaaccess
```

In the appendix, we give documentation on r-nasaaccess conda configuration and installation steps.

### 2.4.3 Tethys

Again, similar to the R software *NASAaccess* package or the conda package (r-nasaaccess) the *NASAaccess* Tethys

application needs a user registration access with Earthdata (https://www.earthdata.nasa.gov/) and storing those credentials in

a the reference file ".netrc" as well as creating a cookies file ".urs_cookies". If the user has successfully prepared the needed

steps to run the *NASAaccess* R package or the conda package (i.e., creating registration access and storing it in a local machine),

then no need to duplicate these steps here again.

The Tethys Platform Framework can be installed in a production or development environment. The difference between a

production and development installation is the development server is not efficient nor capable of handling the traffic a

production   website   receives,   so   a   combination   of   the   NGINX   (https://www.nginx.com/)   and   Daphne

(https://github.com/django/daphne) servers are used for production installations. In addition, when changes are made to a

production installation, such as installing new apps or changing settings, the Daphne server must be restarted manually to load

them. It does not restart automatically like the development server. Usually, the development installation is used for app

development or local use. The Tethys Platform Framework installation process in a development environment is as follows:

1.  Create new conda environment and install the Tethys Platform by running the following command:

```
conda create -n tethys -c tethysplatform -c conda-forge tethys-platform
```





2.  Activate the Tethys conda Environment:

    ```
    conda activate tethys
    ```

3.  Generate a portal_config.yml file containing custom configurations such as the database and other local settings by

    running the following command:

    ```
    tethys gen portal_config
    ```

4.  Tethys Platform requires a PostgreSQL database server. There are several options for setting up a DB server: local,

    docker, or dedicated. The Tethys platform can also be used to create a local server that creates and migrates the tables

associated with the Tethys platform framework by running:

    ```
    tethys db configure
    ```

5.  Finally start the Tethys development server:

    ```
    tethys manage start
    ```

Installation in a production environment can be a manual installation (performing all the production configuration steps

manually) or a docker deployment. The steps for a manual and docker installation can be found in the Tethys platform

documentation (http://docs.tethysplatform.org/en/stable/). Installation of GeoServer is necessary to use the *NASAaccess* Tethys

application. The GeosServer Software can be downloaded and installed on your local machine from https://geoserver.org or

using the Tethys platform, which allows users to pull and run a GeoServer container. The following commands can be used to

install GeoServer through the Tethys Platform, when prompted for settings value, press enter to keep the default values:

```
tethys docker init -c geoserver
tethys docker start -c geoserver
```

If GeoServer was installed from source, start GeoServer by changing into the directory geoserver/bin and executing the

startup.sh script with the following commands:

```
cd geoserver/bin
```
```
sh startup.sh
```

Then, in a web browser, navigate to http://localhost:8080/geoserver to ensure that the GeoServer was installed successfully.

After successful installation of the Tethys Platform and the GeoServer software on your work environment, clone the repository

of the *NASAaccess* application available in Github. Next, install the application into the Tethys platform. Once the installation

has started, the user will be prompted to select a spatial persistent service and the custom settings related to the application.



Finally, start the Tethys development server after the installation has finished. Figure 3 depicts the home window of the

*NASAaccess* Tethys web application. The following commands and steps summarize the process of *NASAaccess* application

installation:

```
1. git clone https://github.com/imohamme/tethys_nasaaccess.git
2. tethys install -d
3. Select the GeoSpatial persistent service (In this case, the installed GeoServer)
4. Enter the value for the custom settings of the NASAaccess application:
   a) data path: custom setting referring to the path of the data directory for download.
   b) nasaaccess_R: custom setting referring to the Rbin path.
   c) nasaaccess_script: custom setting referring to the nasaaccess R script containing the logic for
data download using the r-nasaaccess conda package.
   d) GeoServer workspace: custom setting referring to the GeoServer workspace name associated with
      the NASAacces application.
   e) GeoServer URI: custom setting referring to the GeoServer workspace URI associated with the
      NASAacces application.
f) GeoServer user: custom setting referring to the GeoServer admin user.
   g) GeoServer password: Custom setting referring to the password related to the user of the
      geoserver user setting.
5. Then, starting Tethys
   tethys manage start
```

A detailed installation manual is available at the Github repository of the *NASAaccess* Tethys application

(https://github.com/imohamme/tethys_nasaaccess).

### 2.2 NASAaccess Software Code Availability

All *NASAaccess* related source code and documentation are available online at the following websites:

- *NASAaccess* R package – https://github.com/nasa/NASAaccess,

- *NASAaccess* Python library (`r-nasaaccess`) – https://anaconda.org/conda-forge/r-nasaaccess,

- *NASAaccess* Tethys App – https://github.com/imohamme/tethys_nasaaccess.

The *NASAaccess* code is an open-source NASA Open-Source Agreement v1.3 (https://opensource.org/licenses/NASA-1.3)

and can be downloaded from the above listed sources.

### 3 NASAaccess Implementation

### 3.1 GPM Examples with R and Conda

*NASAaccess* package has multiple functions such as `GPM_NRT`, `GPMpolyCentroid` and `GPMswat` that download, extract,

and reformat rainfall remote sensing data of IMERG from NASA servers (https://gpm.nasa.gov/) for grids within a specified

watershed shapefile. The difference between `GPM_NRT` and `GPMswat` functions is the latency period. The `GPMswat` function



retrieves the IMERG Final Run data which is intended for research quality global multi-satellite precipitation estimates with

quasi-Lagrangian time interpolation, gauge data, and climatological adjustment. On the other hand, the `GPM_NRT` function

retrieves the IMERG near real-time low latency gridded global multi-satellite precipitation estimates.

Let's explore `GPMpolyCentroid` and `GPMswat` functions basic use:

Looking at an example watershed that we want to examine near Houston, Texas in R software platform

```
library(ggmap)
#> Loading required package: ggplot2
#> Google's Terms of Service: https://cloud.google.com/maps-platform/terms/.
#> Please cite ggmap if you use it! See citation("ggmap") for details.
library(raster)
#> Loading required package: sp
library(ggplot2)
library(rgdal)
#> Please note that rgdal will be retired by the end of 2023,
#> plan transition to sf/stars/terra functions using GDAL and PROJ
#> at your earliest convenience.
#>
#> rgdal: version: 1.5-30, (SVN revision 1171)
#> Geospatial Data Abstraction Library extensions to R successfully loaded
#> Loaded GDAL runtime: GDAL 3.4.2, released 2022/03/08
#> Path to GDAL shared files: /Users/imohamme/Library/R/x86_64/4.1/library/rgdal/gdal
#> GDAL binary built with GEOS: FALSE
#> Loaded PROJ runtime: Rel. 8.2.1, January 1st, 2022, [PJ_VERSION: 821]
#> Path to PROJ shared files: /Users/imohamme/Library/R/x86_64/4.1/library/rgdal/proj
#> PROJ CDN enabled: FALSE
#> Linking to sp version:1.4-6
#> To mute warnings of possible GDAL/OSR exportToProj4() degradation,
#> use options("rgdal_show_exportToProj4_warnings"="none") before loading sp or rgdal.
#Reading input data
dem_path <- system.file("extdata",
                        "DEM_TX.tif",
                        package = "NASAaccess")
shape_path <- system.file("extdata",
                          "basin.shp",
                          package = "NASAaccess")
dem <- raster(dem_path)
shape <- readOGR(shape_path)
#> OGR data source with driver: ESRI Shapefile
#> Source:
"/private/var/folders/8t/45w1tdfs1vj3dy1tchbw3pmrhr_gxz/T/Rtmp1IbSo3/temp_libpath3ee86d57d8b5/NASAaccess
/extdata/basin.shp", layer: "basin"
#> with 1 features
#> It has 4 fields
#> Integer64 fields read as strings:  OBJECTID disID
shape.df <- ggplot2::fortify(shape)
#> Regions defined for each Polygons
#plot the watershed data
myMap <- get_stamenmap(bbox = c(left = -96,
                                bottom = 29.7,
                                right = -95.2,
                                top = 30),
                                        maptype = "terrain",
                                        crop = TRUE,
                                         zoom = 10)
#> Source : http://tile.stamen.com/terrain/10/238/422.png
#> Source : http://tile.stamen.com/terrain/10/239/422.png
```



```
#> Source : http://tile.stamen.com/terrain/10/240/422.png
      #> Source : http://tile.stamen.com/terrain/10/241/422.png
      #> Source : http://tile.stamen.com/terrain/10/238/423.png
      #> Source : http://tile.stamen.com/terrain/10/239/423.png
      #> Source : http://tile.stamen.com/terrain/10/240/423.png
#> Source : http://tile.stamen.com/terrain/10/241/423.png
      ggmap(myMap) +
        geom_polygon(data = shape.df,
                     aes(x = long, y = lat, group = group),
                     fill = NA, size = 0.5, color = 'red')
```

Figure 4 depicts the geographic layout of the White Oak Bayou watershed example above. The White Oak Bayou is a tributary

for the Buffalo Bayou River (Harris County, Texas). To use the *NASAaccess* library, we also need a digital elevation model

(DEM) raster layer. The following is an example for the White Oak Bayou watershed DEM and a closer look at the watershed

study example.

```
      # create a plot of our DEM raster along with watershed
library(ggplot2)
      library(raster)
      library(rgdal)
      library(tidyr)
      library(cowplot)
library(ggspatial)
      dem.df <- as.data.frame(dem,xy=TRUE)%>%drop_na()
      ggplot()+
        geom_raster(data=dem.df,aes(x = x,y = y,fill = DEM_TX)) +
        scale_fill_gradientn(name='Elevation (m)', colours = terrain.colors(1000))+
geom_polygon(data = shape.df,aes(x = long, y = lat, group = group),
                   fill = NA, linewidth = 0.5, color = 'black')+
        labs(x='Longitude',y='Latitude')+
        cowplot::theme_cowplot()+
        annotation_north_arrow(location = 'tr', which_north = 'true', pad_x = unit(0.3, 'in'), pad_y =
unit(0.4, 'in'), style = north_arrow_fancy_orienteering(text_size = 8), height =
          unit(0.75, "cm"), width = unit(0.75, "cm")) +
        annotation_scale(plot_unit='km',location = 'tr', width_hint = 0.3, pad_y = unit(0.2, 'in'), pad_x =
                   unit(0.2, 'in'), line_width = 0.8)+
        theme(plot.background = element_rect(color = 1,linewidth = 1),
plot.margin=margin(t = 10, r = 15, b = 10, l = 10, unit = "pt"))
```

Figure 5 gives the White Oak Bayou watershed DEM with elevation range from zero to 50 meters above sea level. After

examining the study watershed and the digital elevation model for it, we can then examine the GPMswat function.

```
      library(NASAaccess)
      GPMswat(Dir = "./GPMswat/",
watershed = shape_path,
                    DEM = dem_path,
                    start = "2020-08-1",
                    end = "2020-08-3")
```

The GPMswat function generated data generated files and a rainfall station file and stored them in the specified Dir.

Examining the rainfall station file generated by GPMswat.

```
      GPMswat.precipitationMaster <- system.file('extdata/GPMswat',
                                        'precipitationMaster.txt',
```



```
                                    package = 'NASAaccess')
     #Reading GPMswat header file
GPMswat.table<-read.csv(GPMswat.precipitationMaster)
     head(GPMswat.table)
     #>       ID            NAME      LAT     LONG ELEVATION
     #> 1 2160842 precipitation2160842 29.93337 -95.82337  50.16166
     #> 2 2160843 precipitation2160843 29.93337 -95.72340  46.68206
#> 3 2160844 precipitation2160844 29.93337 -95.62343  39.72196
     #> 4 2160845 precipitation2160845 29.93337 -95.52346  35.58193
     #> 5 2164442 precipitation2164442 29.83343 -95.82337  48.02116
     #> 6 2164443 precipitation2164443 29.83343 -95.72340  40.47534
     dim(GPMswat.table)
#> [1] 11   5
```

The GPMswat function generated an ascii table for each available grid located within the study watershed. There are 11 grids

within the study watershed and that means 11 tables have been generated. The GPMswat function also generated the rainfall

stations file input shown above GPMswat.table (table with columns: ID, File NAME, LAT, LONG, and ELEVATION)

for those selected grids that fall within the specified watershed. Now, let's see the location of these generated grid points:

```
ggplot() +
       geom_polygon(data = shape.df,
             aes(x = long, y = lat, group = group),
             fill = NA,
             colour = 'black') +
geom_point(data=GPMswat.table,
               aes(x=LONG,
                   y=LAT,
                   fill=ELEVATION),
               shape=21,
size = 4) +
       scale_fill_gradientn(name='Elevation (m)', colours = terrain.colors(7)) +
       labs(x='Longitude',y='Latitude')+
       theme(plot.background = element_rect(color = 1,linewidth = 1),
           plot.margin=margin(t = 10, r = 15, b = 10, l = 10, unit = "pt"))
```

We note here that GPMswat has given us all the GPM data grids that fall within the boundaries of the White Oak Bayou study

watershed (Figure 6). The time series rainfall data stored in the data tables (i.e., 11 tables) can be viewed also by looking at

the reformatted data from the first grid point as listed in the rainfall station file generated by GPMswat.

```
     GPMswat.point.data <- system.file('extdata/GPMswat',
                                'precipitation2160842.txt',
package = 'NASAaccess')
     #Reading data records
     read.csv(GPMswat.point.data)
     #>     X20200801
     #> 1 32.22795868
#> 2  1.80884695
     #> 3  0.07029478
```

The GPMswat has generated a ready format ascii tables that can be ingested easily to any hydrological model of choice.

Now, let's examine GPMpolyCentroid:

```
GPMpolyCentroid(Dir = "./GPMpolyCentroid/",
```



```
watershed = shape_path,
                             DEM = dem_path,
                             start = "2019-08-1",
                             end = "2019-08-3")
```

Examining the rainfall station file generated by `GPMpolyCentroid`:

```
GPMpolyCentroid.precipitationMaster <- system.file('extdata/GPMpolyCentroid',
                                                  'precipitationMaster.txt',
                                                  package = 'NASAaccess')
     GPMpolyCentroid.precipitation.table <- read.csv(GPMpolyCentroid.precipitationMaster)
     #plotting
ggplot() +
       geom_polygon(data = shape.df,
                    aes(x = long, y = lat, group = group),
                    fill = NA,
                    colour = 'red') +
geom_point(data=GPMpolyCentroid.precipitation.table,
                  aes(x=LONG,y=LAT)) +
       labs(x='Longitude',y='Latitude')+
       theme(plot.background = element_rect(color = 1,linewidth = 1),
             plot.margin=margin(t = 10, r = 15, b = 10, l = 10, unit = "pt"))
```

We note here that `GPMpolyCentroid` has given us the GPM data grid that falls within a specified watershed and assigns a

pseudo rainfall gauge located at the centroid of the watershed a weighted-average daily rainfall data (Figure 7). Let's then

examine the precipitation data just obtained by `GPMpolyCentroid` over the White Oak Bayou study watershed.

```
     GPMpolyCentroid.precipitation.record <- system.file('extdata/GPMpolyCentroid',
                                                  'precipitation1.txt',
package = 'NASAaccess')
     GPMpolyCentroid.precipitation.data <- read.csv(GPMpolyCentroid.precipitation.record)
     #since data started on 2019-08-01
     days <- seq.Date(from = as.Date('2019-08-01'),
                      length.out = dim(GPMpolyCentroid.precipitation.data)[1], by = 'day')
#plotting the precipitation time series
     df <- data.frame(day=days,Precipitation=GPMpolyCentroid.precipitation.data [,1])
     ggplot(data=df, aes(days, Precipitation)) +
       geom_point()+
       geom_line()+
labs(x='Longitude',y='Latitude')+
       theme(plot.background = element_rect(color = 1,linewidth = 1),
             plot.margin=margin(t = 10, r = 15, b = 10, l = 10, unit = "pt"))
```

The time series plot above gives the rainfall amounts in (mm) at the centroid of the White Oak Bayou watershed during 2019-

August-01 to 2019-August-03 is shown in Figure 8. Finally, let's examine the near real time precipitation data obtained by

`GPM_NRT` over the White Oak Bayou study watershed. Remember that the minimum latency for `GPM_NRT` is one day.

```
     GPM_NRT(Dir = "./GPMswat/",
                   watershed = shape_path,
                   DEM = dem_path,
                   start = "2022-07-1",
end = "2022-07-3")
```

Let's see the one point data record. See that the data starts on July 1, 2022, and ends on July 3rd, 2022.



```
GPM_NRT.point.data <- system.file('extdata/GPM_NRT',
                                  'precipitation2160845.txt',
                                  package = 'NASAaccess')
#Reading data records
read.csv(GPM_NRT.point.data)
#>   X20220701
#> 1  2.507078
#> 2  1.148573
#> 3  0.000000
```

The above examples were obtained using R version 4.2.2 (R Development Core Team, 2022). The R software program and all

packages used are available from the Comprehensive R Archive Network (CRAN) at https://CRAN.R-project.org. The reader

is encouraged to visit https://imohamme.github.io/NASAaccess/articles/About.html for detailed package documentation and

vignettes including demonstration on GLDAS, CMIP5 & CMIP6. The above *NASAaccess* GPM examples can be easily

replicated in the conda environment by writing the *NASAaccess* commands shown above to a separate file (e.g., `work.R`) and

running it by calling the `Rscript` executable in conda.

In conda, assuming `r-nasaaccess` has been installed successfully, this can be done as:

```
Rscript work.R
```

### 3.2 NASAaccess Tethys Examples

The *NASAaccess* Tethys application adds visualization features to *NASAaccess* R and conda packages. Figure 9 depicts rainfall

remote sensing data of IMERG from NASA servers (https://gpm.nasa.gov/) for grids within the White Oak Bayou watershed

during 2020-January-01 to 2020-December-31 as processed by the `GPMpolyCentroid` function part of the *NASAaccess*

Tethys application. The user can inspect individual grid time series data. This is helpful when looking at different datasets

such as historical and projected air temperature and precipitation time series data at one grid. In Figure 10, we present daily

diurnal air temperature data processed over the same watershed discussed in Figure 9 (the White Oak Bayou watershed) during

the same period (e.g., January 2020 to December 2020). The `GLDASpolyCentroid` function was selected to visualize and

reformat the  Global Land Data Assimilation System GLDAS Noah Land Surface Model L4 3 hourly 0.25 x 0.25 degree V2.1

air temperature dataset (Rodell et al., 2004) in Figure 10.

The *NASAaccess* Tethys application has visualization features for downscaled climate data that includes the CMIP5 & CMIP6

collections. In Figure 11, we give downscaled precipitation data scenario during the year 2045 for the LaPlata Basin derived

from the National Oceanic and Atmospheric Administration (NOAA) Geophysical Fluid Dynamics Laboratory General



Circulation Model (GCM) - GFDL-ESM2M – across the greenhouse gas emission/Representative Concentration Pathways (RCP - rcp85) using the `NEX_GDDP_CMIP5` function. More details on the `NEX_GDDP_CMIP6` & `NEX_GDDP_CMIP5` functions and the downscaled models covered are provided in the appendix - *NASAaccess* documentation.

## 4 Discussion

The *NASAaccess* package presented provides an open-source remote sensing earth observation data access, visualization, and reformat for easy ingestion platform. The biggest advantage we see is the utility of *NASAaccess* in facilitating the access, processing, and visualization various remote sensing earth observation data to scientific and decision maker audiences. This is in-line with the NASA Open-Source Science Initiative (OSSI) call on more open-source science work. This *NASAaccess* work has the potential to increase the remote sensing earth observation data products accessibility on various computing platforms to enhance the progress of science in earth observation data access and management. *NASAaccess* development is in-line with international calls and efforts for open science, scientific information, knowledge, data, and protocols sharing (https://www.unesco.org/en/open-science). We have demonstrated the linkage of *NASAaccess* platform in the SWATOnline example (McDonald et al., 2019) where a decision support system for the lower Mekong River Basin has been shown. Another potential application could be also shown in disseminating climate information for developing countries (Dinku et al., 2014;Dinku et al., 2018) similar to our demonstration in the Se Kong, Se San, and Sre Pok part of the lower Mekong (Mohammed et al., 2022). *NASAaccess* also gives the user automatic, quick, and accurate way for working with remote sensing earth observation data using R and conda environments. This presented application would increase awareness, accelerate progress, and facilitate gaining access to remote sensing earth observation data, tools, and knowledge about our changing environment, moreover it helps to assist in addressing major research gaps in climatological and hydrological science especially in management, interdisciplinary communication, as well as modeling and monitoring.

*NASAaccess* has been introduced to SERVIR (https://www.nasa.gov/mission_pages/servir/overview.html) and GEOGloWS (https://www.geoglows.org/) research network communities through workshops, seminars, and training events. SERVIR, a United States Agency for International Development (USAID) and National Aeronautics and Space Administration (NASA) collaborative project, has multiple global networks that cover different geographic regions such as Hindu-Kush Himalaya, Lower Mekong and Amazonia. For instance, in alignment with the U.S. Indo-Pacific Vision to improve the management of

natural resources SERVIR-Mekong launched a series of regional tools and services utilizing publicly available satellite imagery and geospatial technologies to support the Lower Mekong region to manage environmental risks in enhancing drought resilience and crop yield security, improving regional land cover monitoring, and supporting better flood forecasting and early

warning.

The *NASAaccess* has been also leveraged via GEOGloWS Tethys Portal. The Group of Earth Observation Global Water Sustainability (GEOGloWS) is a voluntary partnership of governments and international organizations. The GEOGloWS provides a framework within which these partners can develop new projects and coordinate their strategies and investments. The GEOGloWS working group 2 Initiative works on the application of information and communication technologies (ICTs),

also known as hydroinformatics, to address the issues related to data analysis, data handling, data management, and data integration methodologies to translate scientific data to knowledge products that are informative, intuitive, understandable, and supportive in the decision-making process. It is important to highlight here that the GEOGloWS Tethys Portal system is free, available for use in location worldwide and developed from services that allow customization for a variety of derivative applications.

In summary, the approach we implemented lowers the barrier between water resources and remote sensing web development as highlighted by Swain et al. (2016). The *NASAaccess* web-based application has visualization capabilities that make it easy to inspect and analyze various remote sensing earth observation data products. Examples of applications of the GPM functions within the platform have been shown. The *NASAaccess* has the advantage that remote sensing data products are easily processed and analyzed within multiple computational frameworks such conda & R. This feature allows users to save the time

for more in-depth analysis. For instance, modelers who are interested in forcing hydrological models with GPM precipitation data will find it very easy to obtain and process GPM data products using *NASAaccess*. In further updates of the platform more earth observation remote sensing products (e.g., ICESat-2 products https://icesat-2.gsfc.nasa.gov/science/data-products) will be implemented to widen the *NASAaccess* utility application areas. Moreover, accessing remote sensing products that characterize water storage changes in lakes, reservoirs, and large river channels obtained through the Surface Water and Ocean

Topography (SWOT) satellite mission (https://swot.jpl.nasa.gov/) will be included.



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

**Acknowledgement**

This work was supported in part by National Aeronautics and Space Administration (NASA) Applied Sciences Grant# NNX16AT88G, and Grant # NNX16AT86G. Any opinions, findings, and conclusions or recommendations expressed in this work are those of the author(s) and do not necessarily reflect the views of the National Aeronautics and Space Agency, Brigham

Young University, Johns Hopkins University, and Science Applications International Corporation.

**Author contribution**

INM conceptualized, developed, and tested the *NASAaccess* R and conda software; GR and INM designed, developed, and tested the *NASAaccess* Tethys web-based application software; INM wrote the manuscript draft; GR, JB, and EJN reviewed and edited the manuscript.

**Competing Interests statement**

The authors declare no competing interests.

**Appendix**

**Conda Installation Documentation**

The `r-nasaaccess` conda package needs user registration with Earthdata (https://www.earthdata.nasa.gov/). As we

discussed earlier in the *NASAaccess* installation steps, users should create a reference file (.netrc) with Earthdata credentials stored in it to streamline the retrieval access to NASA servers. In conda, users should make sure to update conda initial script with '.netrc' file location. Here is the information from a local machine `r-nasaaccess` installation.

```
conda info
        active environment : None
```



```
user config file : /Users/imohamme/.condarc
           populated config files : /Users/imohamme/.condarc
           conda version : 23.1.0
           conda-build version : not installed
           python version : 3.7.12.final.0
virtual packages : __archspec=1=x86_64
                              __osx=10.16=0
                              __unix=0=0
           base environment : /Users/imohamme/opt/miniconda3  (writable)
           conda av data dir : /Users/imohamme/opt/miniconda3/etc/conda
conda av metadata url : None
           channel URLs : https://conda.anaconda.org/conda-forge/osx-64
                          https://conda.anaconda.org/conda-forge/noarch
                          https://conda.anaconda.org/bioconda/osx-64
                          https://conda.anaconda.org/bioconda/noarch
https://conda.anaconda.org/r/osx-64
                          https://conda.anaconda.org/r/noarch
                          https://repo.anaconda.com/pkgs/main/osx-64
                          https://repo.anaconda.com/pkgs/main/noarch
                          https://repo.anaconda.com/pkgs/r/osx-64
https://repo.anaconda.com/pkgs/r/noarch
           package cache : /Users/imohamme/opt/miniconda3/pkgs
                           /Users/imohamme/.conda/pkgs
           envs directories : /Users/imohamme/opt/miniconda3/envs
                              /Users/imohamme/.conda/envs
platform : osx-64
               user-agent : conda/23.1.0 requests/2.28.2 CPython/3.7.12 Darwin/21.6.0 OSX/10.16
               UID:GID : 562380735:1286109195
               netrc file : /Users/imohamme/.netrc
           offline mode : False
```

Installing the `r-nasaaccess` conda package is obtained by:

```
           conda install -c conda-forge r-nasaaccess
```

**NASAaccess Documentation**

The *NASAaccess* documentation contains the following functions:

❖ `NEX_GDDP_CMIP6`

NEX-GDDP-CMIP6 dataset is comprised of downscaled climate scenarios for the globe that are derived from the General

Circulation Model GCM runs conducted under the Coupled Model Intercomparison Project Phase 6 CMIP6 (Eyring et al.,

2016) and across the four "Tier 1" greenhouse gas emissions scenarios known as Shared Socioeconomic Pathways SSPs

(O'Neill et al., 2016;Meinshausen et al., 2020). The CMIP6 GCM runs were developed in support of the Sixth Assessment

Report of the Intergovernmental Panel on Climate Change IPCC AR6. This data set includes downscaled projections from the

35 models and scenarios for which daily scenarios were produced and distributed under CMIP6. The Bias-Correction Spatial

Disaggregation BCSD method used in generating the NEX-GDDP-CMIP6 data set is a statistical downscaling algorithm

specifically developed to address the current limitations of the global GCM outputs (Wood et al., 2002;Wood et al.,





2004;Maurer and Hidalgo, 2008;Thrasher et al., 2012). The NEX-GDDP-CMIP6 climate projections is downscaled at a spatial resolution of 0.25 degrees x 0.25 degrees (approximately 25 km x 25 km). The `NEX_GDDP_CMIP6` downscales the NEX-

GDDP data to grid points of 0.1 degrees x 0.1 degrees following nearest point methods described by Mohammed et al. (2018). The `NEX_GDDP_CMIP6` syntax is as follows:

```
NEX_GDDP_CMIP6(Dir = "./INPUT/", watershed = "watershed.shp", DEM = "watershed_dem.tif", start = "2060-
12-1", end = "2060-12-3", model ="MIROC6" , type ="pr" , slice ="ssp245")                                (7)
```

Arguments:

`Dir`

A directory name to store gridded climate data and stations files,

`watershed`

A study watershed shapefile spatially describing polygon(s) in a geographic projection sp::CRS('+proj=longlat +datum=WGS84'),

`DEM`

A study watershed digital elevation model raster in a geographic projection sp::CRS('+proj=longlat +datum=WGS84'),

`start`

Beginning date for gridded climate data, and it should be equal to or greater than 2006-Jan-01 for 'rcp45' or 'rcp85' RCP climate scenario. Also, `start` should be equal to or greater than 1950-Jan-01 and `end` should be equal to or less than 2005-Dec-31

for the 'historical' GCM retrospective climate data.

`end`

Ending date for gridded climate data.

`model`

A climate modeling center and name from the World Climate Research Programme WCRP global climate projections through

the Coupled Model Intercomparison Project 6 CMIP6 (e.g., MIROC6 which is the sixth version of the Model for Interdisciplinary Research on Climate MIROC model).

`type`

A flux data type. It's value can be 'pr' for precipitation or 'tas' for air temperature.



```
slice
```

A scenario from the Shared Socioeconomic Pathways (SSPs). It's value can be 'ssp126', 'ssp245', 'ssp370', 'ssp585', or

'historical'.

✦  `NEX_GDDP_CMIP5`

The `NEX_GDDP_CMIP5` function downloads and processes climate change data of rainfall and air temperature from NASA

Earth Exchange Global Daily Downscaled Projections NEX-GDDP GSFC servers ([https://www.nccs.nasa.gov/services/data-](https://www.nccs.nasa.gov/services/data-collections/land-based-products/nex-gddp)

[collections/land-based-products/nex-gddp](https://www.nccs.nasa.gov/services/data-collections/land-based-products/nex-gddp)), extracts data from grids within a specified watershed shapefile, and then generates

tables in a format that any hydrological model requires for rainfall or air temperature data input. The `NEX_GDDP_CMIP5`

function also generates the climate stations file input (file with columns: ID, File NAME, LAT, LONG, and ELEVATION)

for those selected climatological grids that fall within the specified watershed. The NASA Earth Exchange Global Daily

Downscaled Projections NEX-GDDP dataset is comprised of downscaled climate scenarios for the globe that are derived from

the General Circulation Model (GCM) runs conducted under the Coupled Model Intercomparison Project Phase 5 CMIP5 and

across two of the four greenhouse gas emissions scenarios known as Representative Concentration Pathways RCPs (rcp45,

rcp85). The NEX-GDDP dataset is comprised of downscaled climate scenarios for the globe that are derived from the General

Circulation Model GCM runs conducted under the Coupled Model Intercomparison Project Phase 5 CMIP5 (Taylor et al.,

2012) and across two of the four greenhouse gas emissions scenarios known as Representative Concentration Pathways RCPs

(Meinshausen et al., 2011). The CMIP5 GCM runs were developed in support of the Fifth Assessment Report of the

Intergovernmental Panel on Climate Change IPCC AR5. This dataset includes downscaled projections from the 21 models and

scenarios for which daily scenarios were produced and distributed under CMIP5. The Bias-Correction Spatial Disaggregation

BCSD method used in generating the NEX-GDDP dataset is a statistical downscaling algorithm specifically developed to

address the current limitations of the global GCM outputs (Wood et al., 2002;Wood et al., 2004;Maurer and Hidalgo,

2008;Thrasher et al., 2012). The NEX-GDDP climate projections are downscaled at a spatial resolution of 0.25 degrees x 0.25

degrees (approximately 25 km x 25 km). The `NEX_GDDP_CMIP5`  downscales the NEX-GDDP data to grid points of 0.1

degrees x 0.1 degrees following nearest point methods described by Mohammed et al. (2018). The `NEX_GDDP_CMIP5` syntax

is as follows:



```
NEX_GDDP_CMIP5(Dir = "./INPUT/", watershed = "watershed.shp", DEM = "watershed_dem.tif", start = "2060-
12-1", end = "2060-12-3", model ="IPSL-CM5A-MR" , type ="pr" , slice ="rcp85")                    (6)
```

Arguments:

`Dir`

A directory name to store gridded climate data and stations files,

`watershed`

A study watershed shapefile spatially describing polygon(s) in a geographic projection sp::CRS('+proj=longlat +datum=WGS84'),

`DEM`

A study watershed digital elevation model raster in a geographic projection sp::CRS('+proj=longlat +datum=WGS84'),

`start`

Beginning date for gridded climate data, and it should be equal to or greater than 2006-Jan-01 for 'rcp45' or 'rcp85' RCP climate scenario. Also, `start` should be equal to or greater than 1950-Jan-01 and `end` should be equal to or less than 2005-Dec-31 for the 'historical' GCM retrospective climate data.

`end`

Ending date for gridded climate data.

`model`

A climate modeling center and name from the World Climate Research Programme WCRP global climate projections through the Coupled Model Intercomparison Project 5 CMIP5 (e.g., IPSL-CM5A-MR which is Institut Pierre-Simon Laplace CM5A-MR model).

`type`

A flux data type. It's value can be 'pr' for precipitation or 'tas' for air temperature.

`slice`

A scenario from the Representative Concentration Pathways. It's value can be 'rcp45', 'rcp85', or 'historical'.

❖  `GPM_NRT`

The `GPM_NRT` function downloads and processes rainfall remote sensing data of the Integrated Multi-satellitE Retrievals for

GPM (IMERG) from NASA GSFC servers, extracts data from grids within a specified watershed shapefile, and then generates

tables in a format that any hydrological model requires for rainfall data input. The `GPM_NRT` function also generates the

rainfall stations file input (file with columns: ID, File NAME, LAT, LONG, and ELEVATION) for those selected grids that

fall within the specified watershed. The minimum latency for the `GPM_NRT` function is one day. The `GPM_NRT` function

accesses NASA Goddard Space Flight Center server address for IMERG remote sensing data products at

(https://gpm1.gesdisc.eosdis.nasa.gov/data/GPM_L3/GPM_3IMERGDE.06/). The IMERG dataset used by `GPM_NRT` is the

GPM Level 3 IMERG *Early* Daily 0.1 x 0.1 deg (GPM_3IMERGDE) derived from the half-hourly GPM_3IMERGHHE.

The derived result represents the final estimate of the daily accumulated precipitation. The IMERG dataset is produced at the

NASA Goddard Earth Sciences (GES) Data and Information Services Center (DISC) by simply summing the valid

precipitation retrievals for the day in GPM_3IMERGHHE and giving the result in millimeters. The `GPM_NRT` function uses

variable name ('precipitationCal') for rainfall in IMERG data products. The IMERG data products are available from 2000-

June-1 to present. The `GPM_NRT` function outputs table and gridded data files matching grid points resolution of IMERG data

products (i.e., resolution of 0.1 degree). The `GPM_NRT` syntax is as follows:

```
GPM_NRT(Dir = "./INPUT/", watershed = "watershed.shp", DEM = "watershed_dem.tif", start = "2015-12-1",
end = "2015-12-3")                                                                          (3)
```

Arguments:

`Dir`

A directory name to store gridded rainfall and rain stations files,

`watershed`

A study watershed shapefile spatially describing polygon(s) in a geographic projection sp::CRS('+proj=longlat

+datum=WGS84'),

`DEM`

A study watershed digital elevation model raster in a geographic projection sp::CRS('+proj=longlat +datum=WGS84'),

`start`

Beginning date for gridded rainfall data and it should be equal to or greater than 2000-Jun-01,



`end`

Ending date for gridded rainfall data.

❖  `GPMpolyCentroid`

The `GPMpolyCentroid` function downloads and processes rainfall remote sensing data of IMERG from NASA GSFC

servers, extracts data from grids falling within a specified sub-basin(s) watershed shapefile and assigns a pseudo rainfall gauge

located at the centroid of the sub-basin(s) watershed a weighted-average daily rainfall data. The function generates rainfall

tables in a format that any rainfall-runoff hydrological model requires for rainfall data input. The function also generates the

rainfall stations file summary input (file with columns: ID, File NAME, LAT, LONG, and ELEVATION) for those pseudo

grids that correspond to the centroids of the watershed sub-basins. The minimum latency for the `GPMpolyCentroid`

function is 3.5 months. The `GPMpolyCentroid` function accesses NASA Goddard Space Flight Center server address for

IMERG remote sensing data products at (https://gpm1.gesdisc.eosdis.nasa.gov/data/GPM_L3/GPM_3IMERGDF.06/). The

IMERG dataset used by the `GPMpolyCentroid` function is the GPM Level 3 IMERG *Final* Daily 0.1 x 0.1 deg

(GPM_3IMERGDF) derived from the half-hourly GPM_3IMERGHH. This derived result represents the final estimate of the

daily accumulated precipitation. The GPM_3IMERGDF dataset is produced at NASA Goddard Earth Sciences (GES) Data

and Information Services Center (DISC) by simply summing the valid precipitation retrievals for the day in GPM_3IMERGHH

and giving the result in millimetres. The `GPMpolyCentroid` syntax is as follows:

```
GPMpolyCentroid(Dir = "./INPUT/", watershed = "watershed.shp", DEM = "watershed_dem.tif", start = "2015-
12-1", end = "2015-12-3")                                                                      (4)
```

Arguments:

`Dir`

A directory name to store gridded rainfall and rain stations files,

`watershed`

A study watershed shapefile spatially describing polygon(s) in a geographic projection sp::CRS('+proj=longlat

+datum=WGS84'),

`DEM`

A study watershed digital elevation model raster in a geographic projection sp::CRS('+proj=longlat +datum=WGS84'),



start

Beginning date for gridded rainfall data and it should be equal to or greater than 2000-Mar-01,

end

Ending date for gridded rainfall data.

❖  GPMswat

The GPMswat function downloads and processes rainfall remote sensing data of IMERG from NASA GSFC servers, extracts

data from grids within a specified watershed shapefile, and then generates tables in a format that the Soil and Water Assessment

Tool (SWAT) (https://swat.tamu.edu/) hydrological model requires for rainfall data input. The function also generates the

rainfall stations file input (file with columns: ID, File NAME, LAT, LONG, and ELEVATION) for those selected grids that

fall within the specified watershed. The minimum latency for the GPMswat function is 3.5 months. The GPMswat function

accesses NASA Goddard Space Flight Center server address for IMERG remote sensing data products at

(https://gpm1.gesdisc.eosdis.nasa.gov/data/GPM_L3/GPM_3IMERGDF.06/). The IMERG dataset used by the GPMswat

function is the GPM Level 3 IMERG *Final* Daily 0.1 x 0.1 deg (GPM_3IMERGDF) derived from the half-hourly

GPM_3IMERGHH. This derived result represents the final estimate of the daily accumulated precipitation. The

GPM_3IMERGDF dataset is produced at NASA Goddard Earth Sciences (GES) Data and Information Services Center (DISC)

by simply summing the valid precipitation retrievals for the day in GPM_3IMERGHH and giving the result in millimetres.

The GPMswat syntax is as follows:

```
GPMswat(Dir="./INPUT/", watershed = "watershed.shp", DEM = "watershed_dem.tif", start = "2015-12-1", end
        = "2015-12-3")                                                                                    (5)
```

Arguments:

Dir

A directory name to store gridded rainfall and rain stations files,

watershed

A study watershed shapefile spatially describing polygon(s) in a geographic projection sp::CRS('+proj=longlat

+datum=WGS84'),

DEM

A study watershed digital elevation model raster in a geographic projection sp::CRS('+proj=longlat +datum=WGS84'),

`start`

Beginning date for gridded rainfall data and it should be equal to or greater than 2000-Mar-01,

`end`

Ending date for gridded rainfall data.

❖ `GLDASpolyCentroid`

The `GLDASpolyCentroid` function downloads and processes remote sensing data product of GLDAS from NASA Goddard Space Flight Center (GSFC) servers, extracts air temperature data from grids falling within a specified sub-basin(s) watershed

shapefile and assigns a pseudo air temperature gauge located at the centroid of the sub-basin(s) watershed a weighted-average daily minimum and maximum air temperature data. The `GLDASpolyCentroid` function generates ascii tables in a format that any rainfall-runoff hydrological model requires for minimum and maximum air temperatures data input. The `GLDASpolyCentroid` function outputs gridded air temperature data in degree Celsius. The `GLDASpolyCentroid` function also generates air temperature stations file input (file with columns: ID, File NAME, LAT, LONG, and ELEVATION)

for those pseudo grids that correspond to the centroids of the watershed sub-basins. The `GLDASpolyCentroid` syntax is as follows:

```
GLDASpolyCentroid(Dir = "./INPUT/", watershed = "watershed.shp" , DEM = "watershed_dem.tif" , start =
"2015-12-1" , end = "2015-12-3")                                                      (1)
```

Arguments:

`Dir`

A directory name to store gridded air temperature and air temperature stations files,

`watershed`

A study watershed shapefile spatially describing polygon(s) in a geographic projection sp::CRS('+proj=longlat +datum=WGS84'),

`DEM`

A study watershed digital elevation model raster in a geographic projection sp::CRS('+proj=longlat +datum=WGS84'),

`start`

Beginning date for gridded air temperature data and it should be equal to or greater than 2000-Jan-01,

```
end
```

Ending date for gridded air temperature data.

❖  `GLDASwat`

The `GLDASwat` function downloads and processes remote sensing data products of GLDAS from NASA GSFC servers,

extracts air temperature data from grids within a specified watershed shapefile, and then generates tables in a format that the

Soil and Water Assessment Tool (SWAT) (https://swat.tamu.edu/) hydrological model requires for minimum and maximum

air temperature data input. The `GLDASwat` function finds the minimum and maximum air temperatures for each day at each

grid within the study watershed by searching for minima and maxima over the three hours air temperature data values available

for each day and grid. The `GLDASwat` function outputs gridded air temperature data in degree Celsius. The `GLDASwat`

function also generates the air temperature stations file input (file with columns: ID, File NAME, LAT, LONG, and

ELEVATION) for those selected grids that fall within the specified watershed. The `GLDASwat` syntax is as follows:

`GLDASwat(Dir = "./INPUT/", watershed = "watershed.shp", DEM = "watershed_dem.tif", start = "2015-12-1",`

`end = "2015-12-3")`                                                                                   (2)

Arguments:

```
Dir
```

A directory name to store gridded air temperature and air temperature stations files,

`watershed`

A study watershed shapefile spatially describing polygon(s) in a geographic projection sp::CRS('+proj=longlat

+datum=WGS84'),

```
DEM
```

A study watershed digital elevation model raster in a geographic projection sp::CRS('+proj=longlat +datum=WGS84'),

`start`

Beginning date for gridded air temperature data and it should be equal to or greater than 2000-Jan-01,

```
end
```

Ending date for gridded air temperature data.





**Table 1. Selected NASA GES DISC tools and services for accessing and visualizing earth observation remote sensing data**

| Name | Description | Link | Reference |
|---|---|---|---|
| Giovanni | A web application that provides a simple, intuitive way to visualize, analyze, and access Earth science remote sensing data, particularly from satellites, without having to download the data. | https://giovanni.gsfc.nasa.gov/giovanni/ | (Acker and Leptoukh, 2007;Berrick et al., 2009;Teng et al., 2014) |
| Data Quality Visualization (DQViz) | A visualization service supporting various visualization and data accessing capabilities from satellite Level 2 (MODIS/MISR/OMI) and long term assimilated aerosols from NASA Modern-Era Retrospective analysis for Research and Applications. | https://disc1.gesdisc.eosdis.nasa.gov/dqviz/index.htm | (Wei et al., 2016) |
| Hydrology Data Rods | A tool for selected data rods variables that can be reorganized as time series, searched and accessed through the GES DISC search and access user interface. | | (Teng et al., 2016) |
| OGC Web Map Server (WMS) | A service that provides users with geo-registered maps (images) produced from various GES DISC data products. | WMS server for the Atmospheric Infrared Sounder (AIRS) data product:<br><br>https://disc1.gesdisc.eosdis.nasa.gov/daac-bin/wms_airs?service=wms&version=1.1.1&request=getcapabilities<br><br>WMS server for the Ozone Mapping Instrument (OMI) data product: | |





| | | https://disc1.gesdisc.eosdis.nasa.gov/daac-bin/wms_omi?service=wms&version=1.1.1&request=getcapabilities<br><br>WMS server for near real-time Atmospheric Infrared Sounder (AIRS) data product:<br><br>https://disc1.gesdisc.eosdis.nasa.gov/daac-bin/wms_airsnrt?service=wms&version=1.1.1&request=getcapabilities<br><br>WMS server for selected TRMM precipitation data product:<br><br>https://disc1.gesdisc.eosdis.nasa.gov/daac-bin/wms_trmm?service=wms&version=1.1.1&request=getcapabilities<br><br>WMS server for selected GES DISC science data products which are available from the Giovanni data analysis and visualization portal:<br><br>https://giovanni.gsfc.nasa.gov/giovanni/daac-bin/wms_ag4?SERVICE=WMS&VERSION=1.1.1&REQUEST=Getcapabilities | |
| OPenNDAP and GDS | Web Services provides remote access to individual variables within datasets in a form usable by many tools, such as IDV, McIDAS-V, | https://disc.gsfc.nasa.gov/information/tools?title=OPeNDAP%20and%20GDS | |





| | Panoply, Ferret and GrADS, etc. | | |
|---|---|---|---|
| Mirador | A subsetting service that provides on-the-fly parameter and spatial subsetted files. | | (Lynnes et al., 2009) |


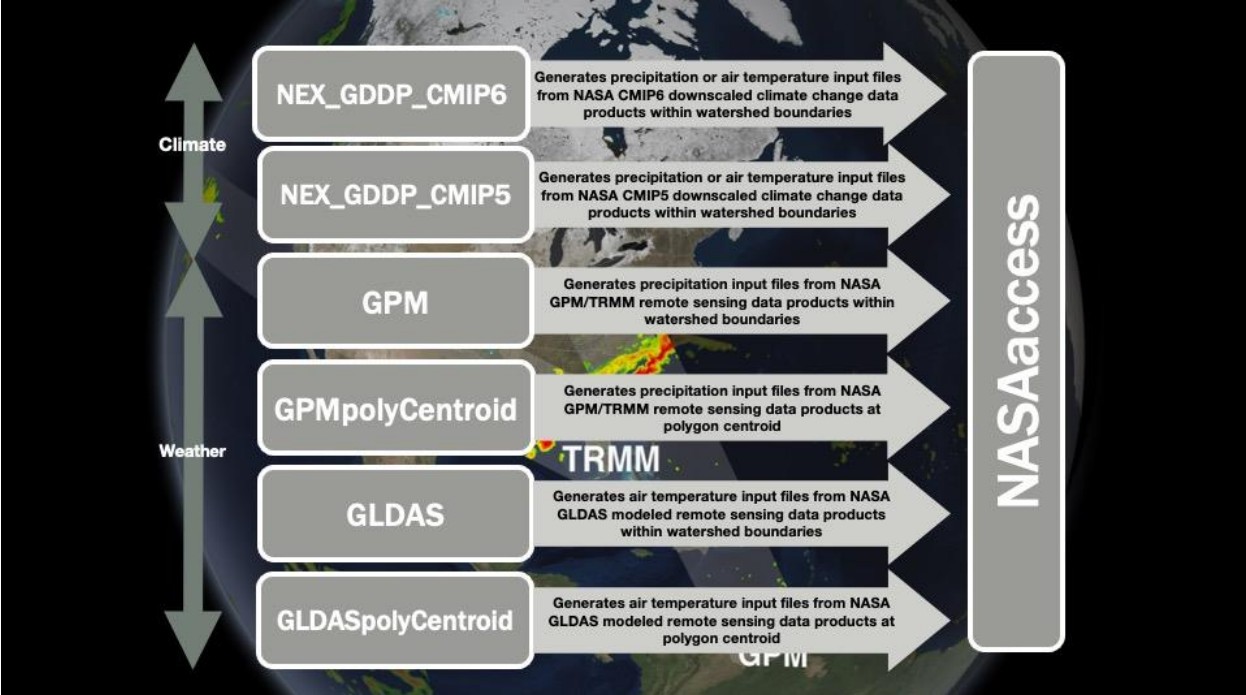

**Figure 1:** *NASAaccess* **available functions (NASAaccess version 3.3.0).**





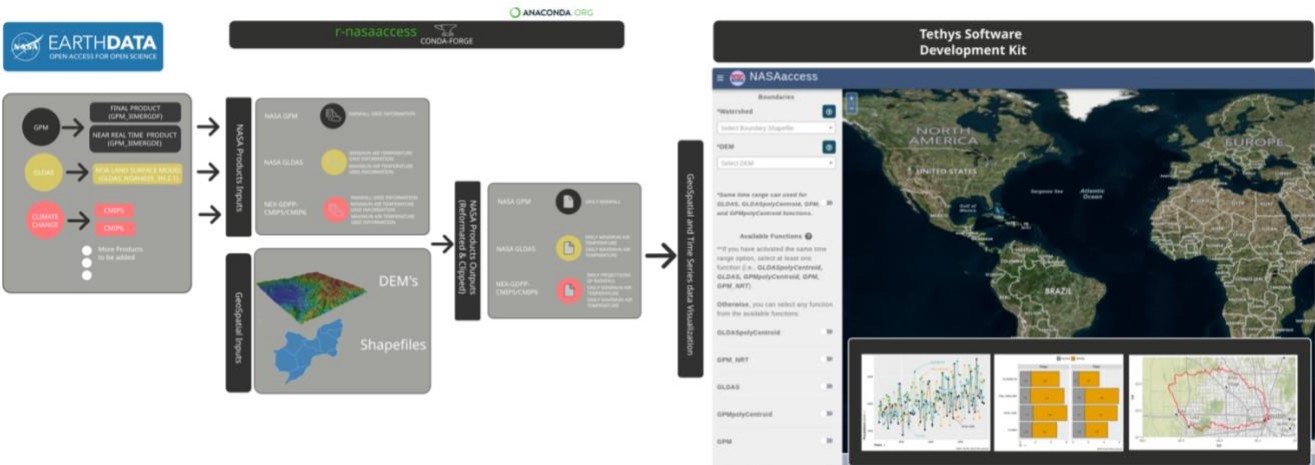

**Figure 2:** *NASAaccess* **Tethys application flow chart.**

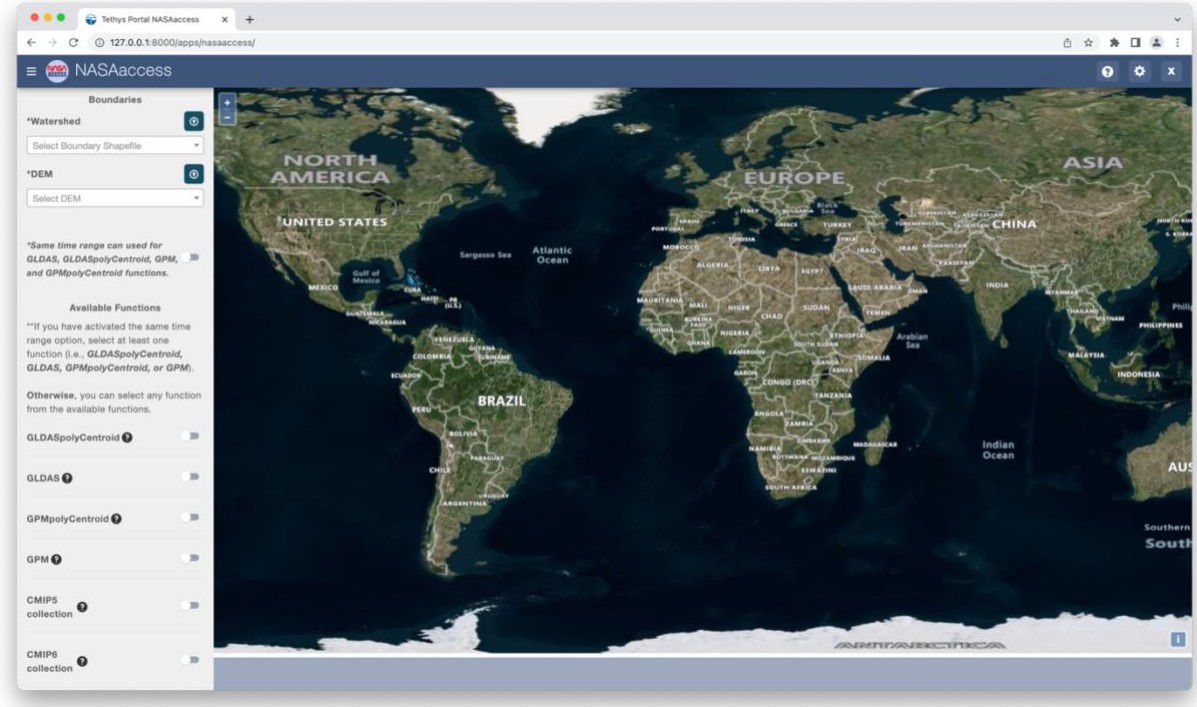

**Figure 3:** *NASAaccess* **Tethys application home window.**



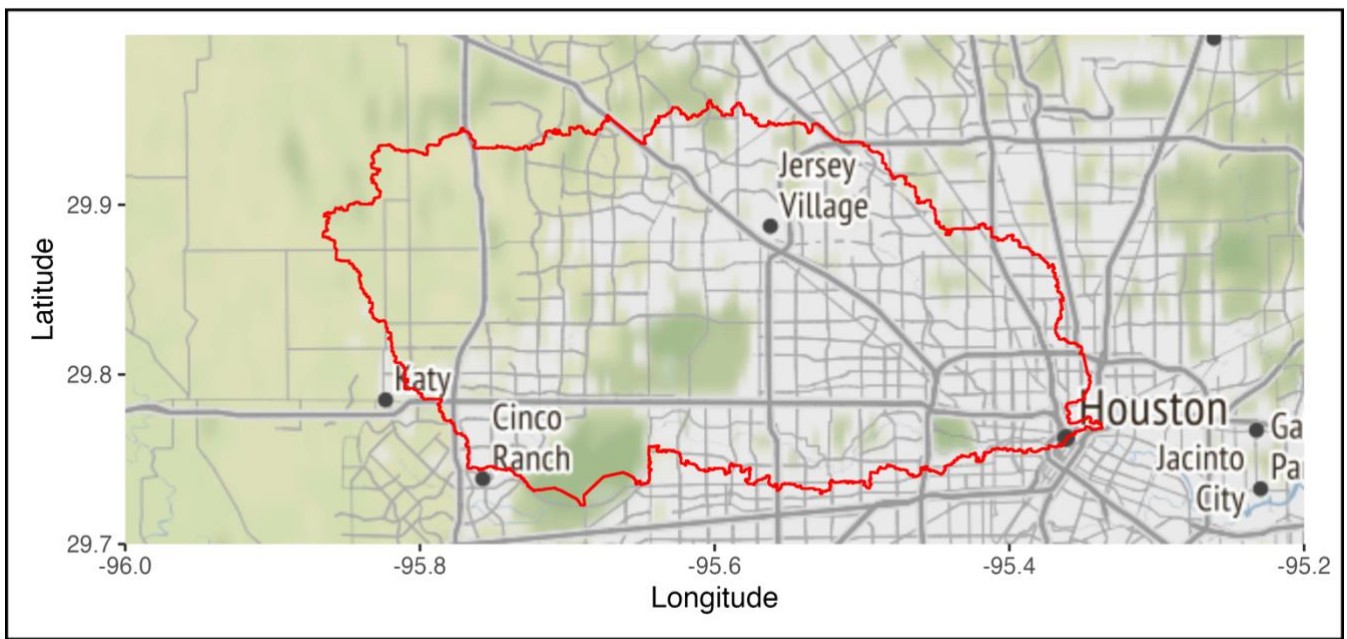

**Figure 4: The geographic layout of the White Oak Bayou watershed. The White Oak Bayou is a tributary for the Buffalo Bayou River (Harris County, Texas). Map created and drafted using R: A language and environment for statistical computing version 4.2.2: https://www.R-project.org/ (Vienna, Austria). The map layout was plotted using EPSG Geodetic Parameter Dataset 4326 projection (https://epsg.io/4326).**

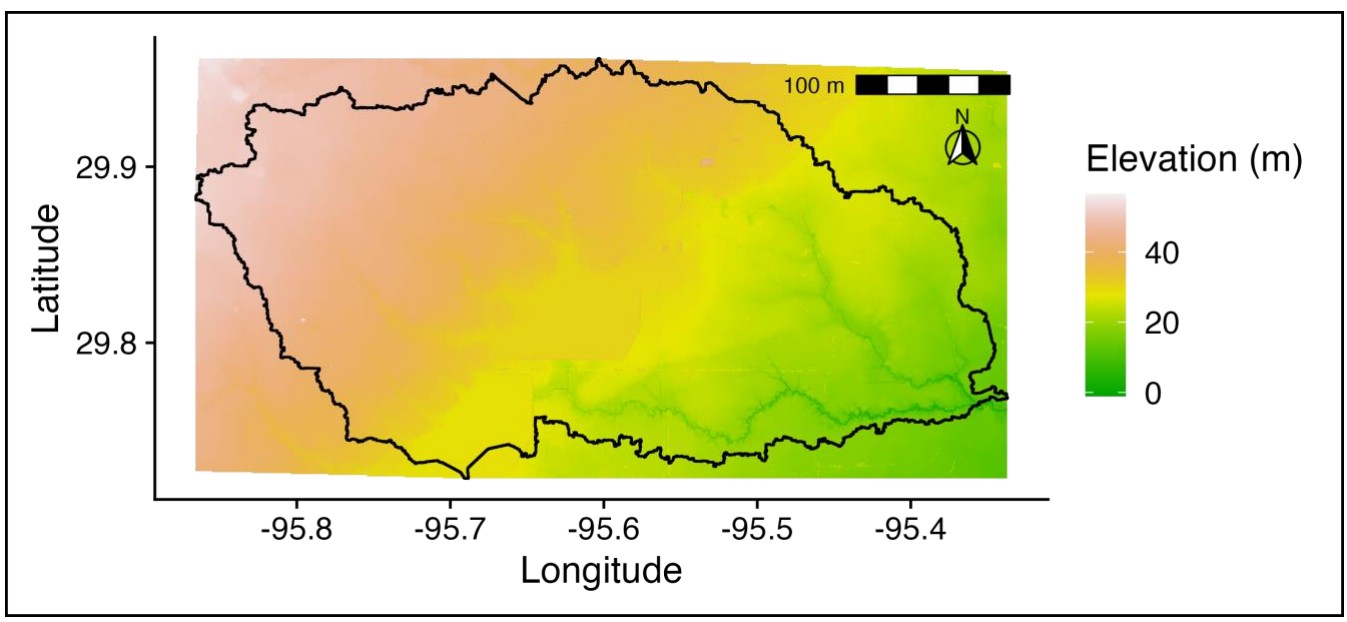



**Figure 5: The White Oak Bayou watershed with digital elevation model. The White Oak Bayou is a tributary for the Buffalo Bayou River (Harris County, Texas). Map created and drafted using R: A language and environment for statistical computing version 4.2.2: https://www.R-project.org/ (Vienna, Austria). The map layout was plotted using EPSG Geodetic Parameter Dataset 4326 projection (https://epsg.io/4326).**

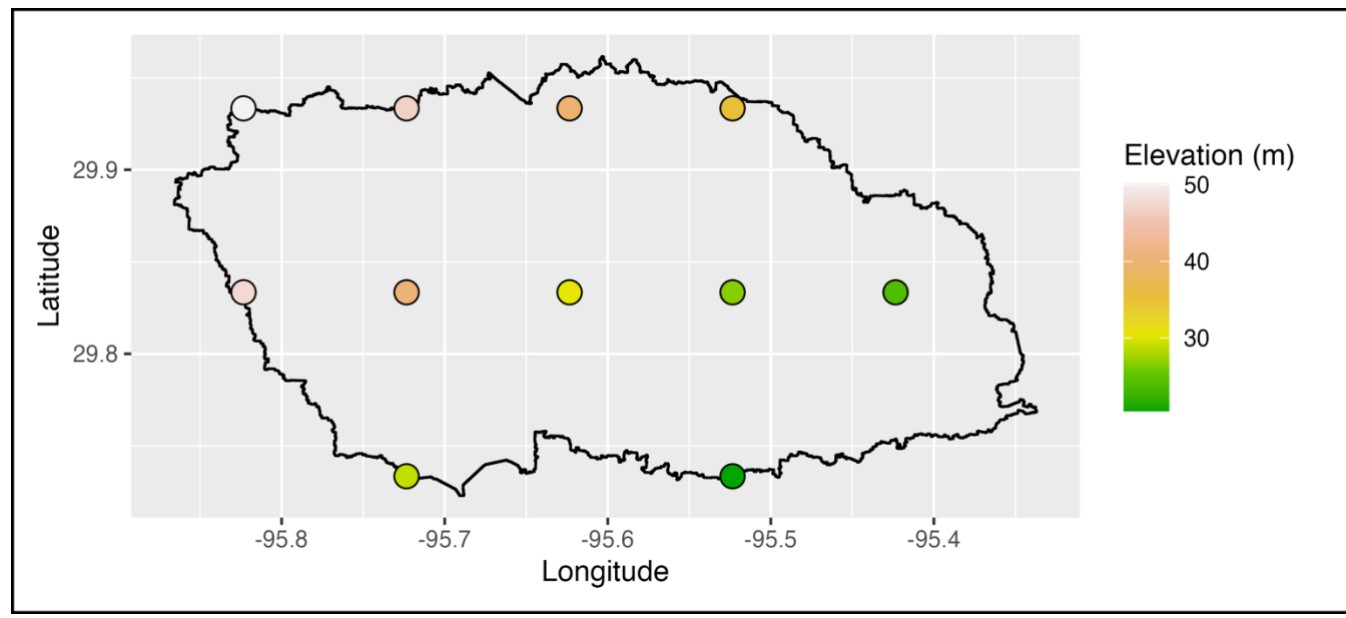

**Figure 6: The geographic layout of the White Oak Bayou watershed with all IMERG dataset (GPM Level 3 IMERG *Final* Daily 0.1 x 0.1 deg, GPM_3IMERGDF) derived from the half-hourly GPM_3IMERGHH data product grids obtained by the `GPMswat` function of the *NASAaccess* package that fall within the watershed boundaries. The White Oak Bayou is a tributary for the Buffalo Bayou River (Harris County, Texas). Map created and drafted using R: A language and environment for statistical computing version 4.2.2: https://www.R-project.org/ (Vienna, Austria). The map layout was plotted using EPSG Geodetic Parameter Dataset 4326 projection (https://epsg.io/4326).**

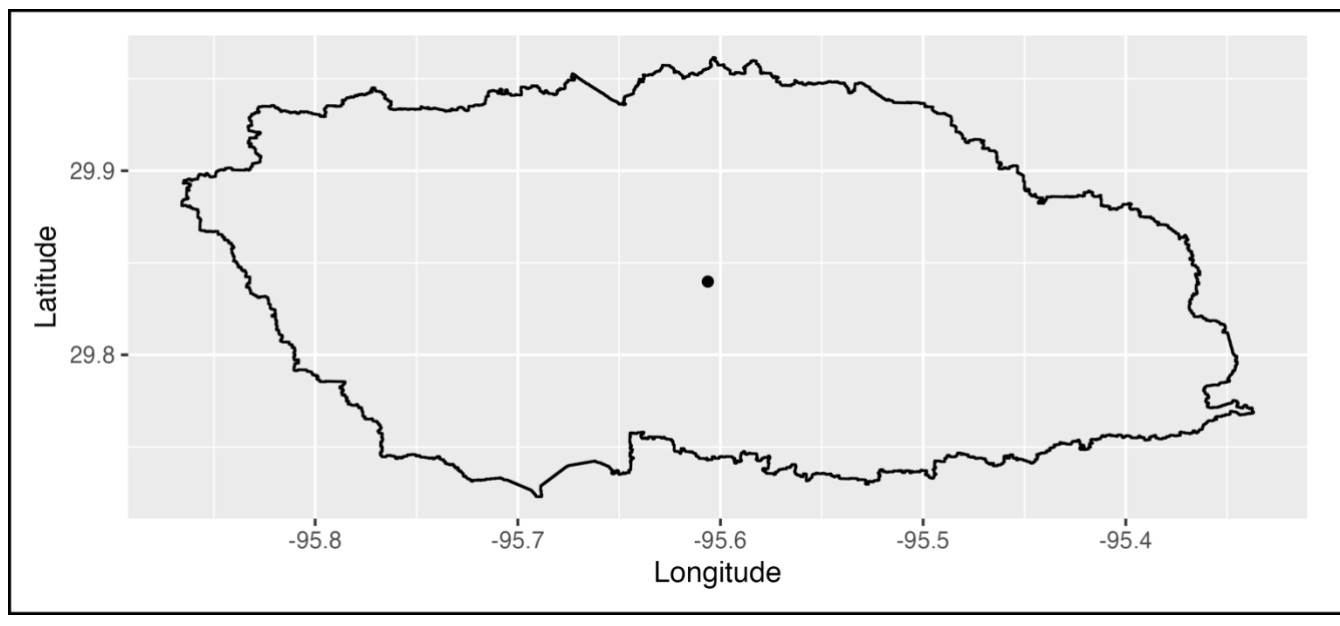




**Figure 7: The geographic layout of the White Oak Bayou watershed with a data grid obtained by the `GPMpolyCentroid` function of the *NASAaccess* package. The `GPMpolyCentroid` function assigns a pseudo rainfall gauge located at the centroid of the watershed a weighted-average daily rainfall data from IMERG dataset (GPM Level 3 IMERG \*Final\* Daily 0.1 x 0.1 deg, GPM_3IMERGDF) derived from the half-hourly GPM_3IMERGHH data products. The White Oak Bayou is a tributary for the**
950 **Buffalo Bayou River (Harris County, Texas). Map created and drafted using R: A language and environment for statistical computing version 4.2.2: https://www.R-project.org/ (Vienna, Austria). The map layout was plotted using EPSG Geodetic Parameter Dataset 4326 projection (https://epsg.io/4326).**

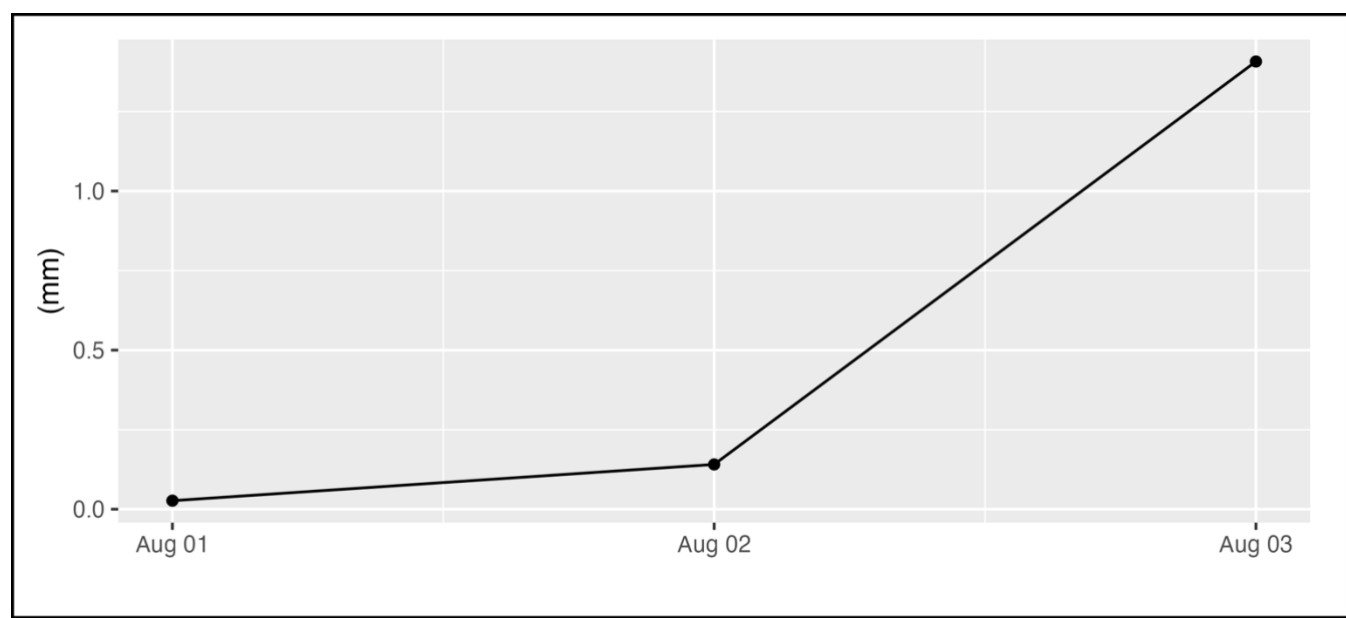

**Figure 8: The rainfall amounts in (mm) at the centroid of the White Oak Bayou watershed from 2019-August-01 to 2019-August-03**
**as obtained by the `GPMpolyCentroid` function. The `GPMpolyCentroid` function assigns a pseudo rainfall gauge located at the centroid of the watershed a weighted-average daily rainfall data from IMERG dataset (GPM Level 3 IMERG \*Final\* Daily 0.1 x 0.1 deg, GPM_3IMERGDF) derived from the half-hourly GPM_3IMERGHH data products. The White Oak Bayou is a tributary for the Buffalo Bayou River (Harris County, Texas).**



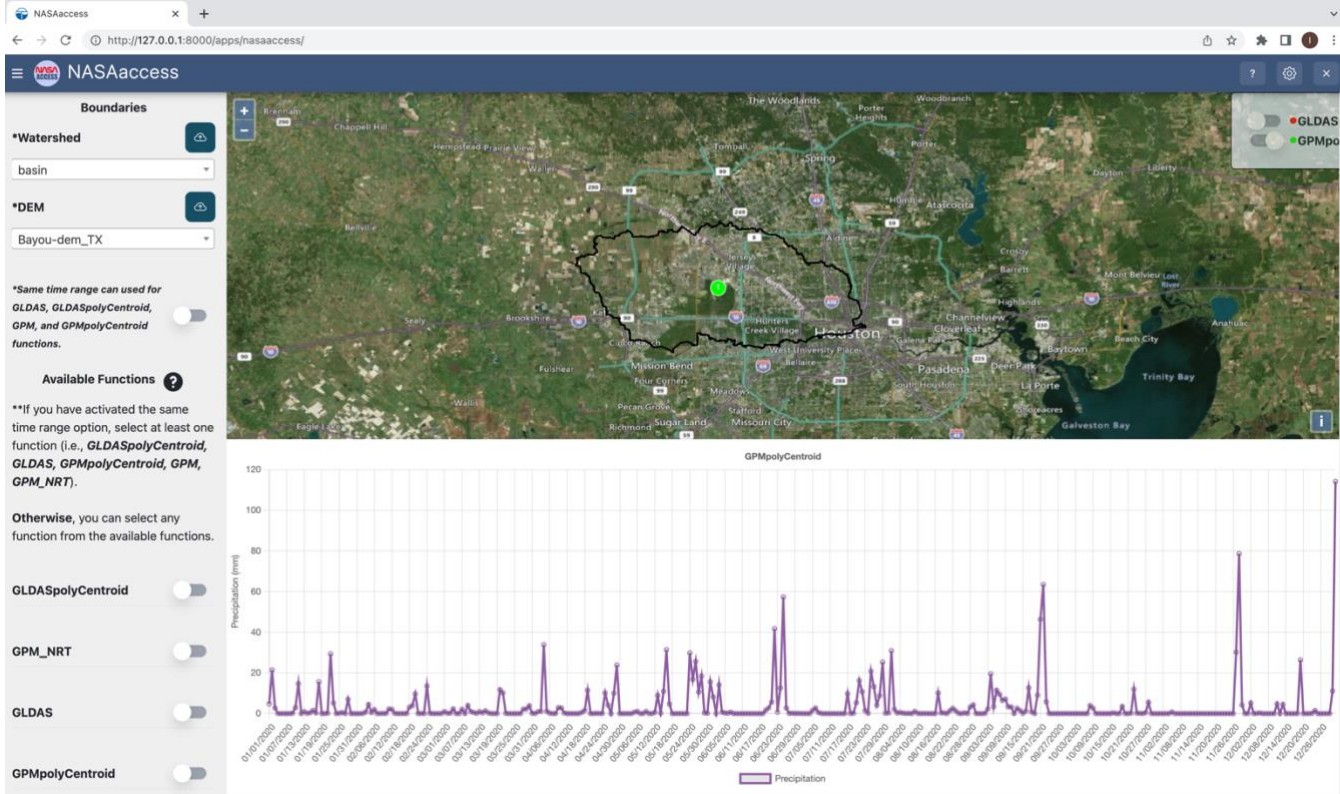

**Figure 9: The rainfall amounts in (mm) at the centroid of the White Oak Bayou watershed from 2020-January-01 to 2020-December-31 as obtained by the `GPMpolyCentroid` function and presented by the *NASAaccess* Tethys application. The `GPMpolyCentroid` function assigns a pseudo rainfall gauge located at the centroid of the watershed a weighted-average daily rainfall data from IMERG dataset (GPM Level 3 IMERG *Final* Daily 0.1 x 0.1 deg, GPM_3IMERGDF) derived from the half-hourly GPM_3IMERGHH data products. The White Oak Bayou is a tributary for the Buffalo Bayou River (Harris County, Texas). Map created and drafted using © Microsoft Bing Maps Platform API.**



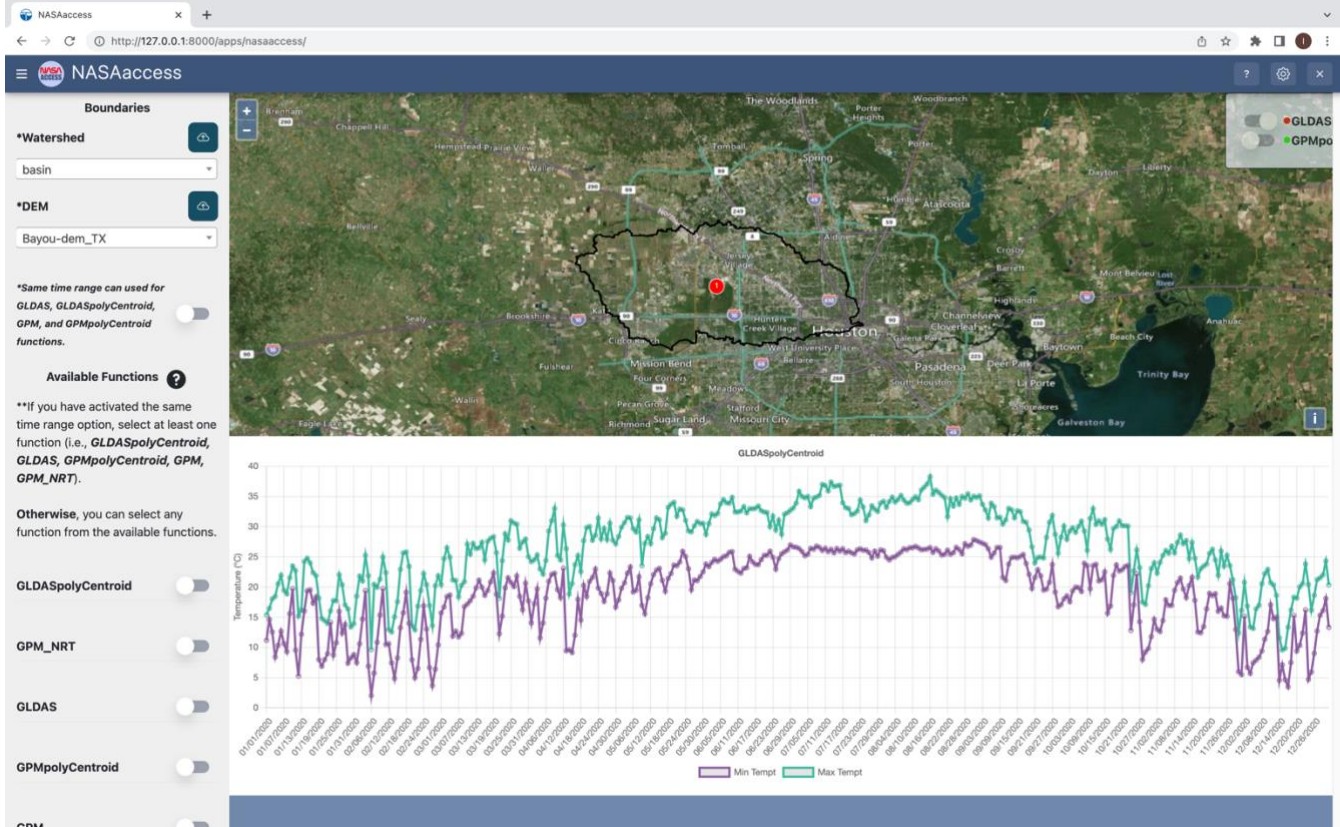

**Figure 10: The daily diurnal air temperature (minimum and maximum) in degree Celsius at the centroid of the White Oak Bayou watershed from 2020-January-01 to 2020-December-31 as obtained by the `GLDASpolyCentroid` function of the *NASAaccess* Tethys application. The `GLDASpolyCentroid` function assigns a pseudo air temperature gauge located at the centroid of the White Oak Bayou watershed a weighted-average daily minimum and maximum air temperature data from the GLDAS Noah Land Surface Model L4 3 hourly 0.25 x 0.25 degree V2.1. The White Oak Bayou is a tributary for the Buffalo Bayou River (Harris County, Texas). Map created and drafted using © Microsoft Bing Maps Platform API.**



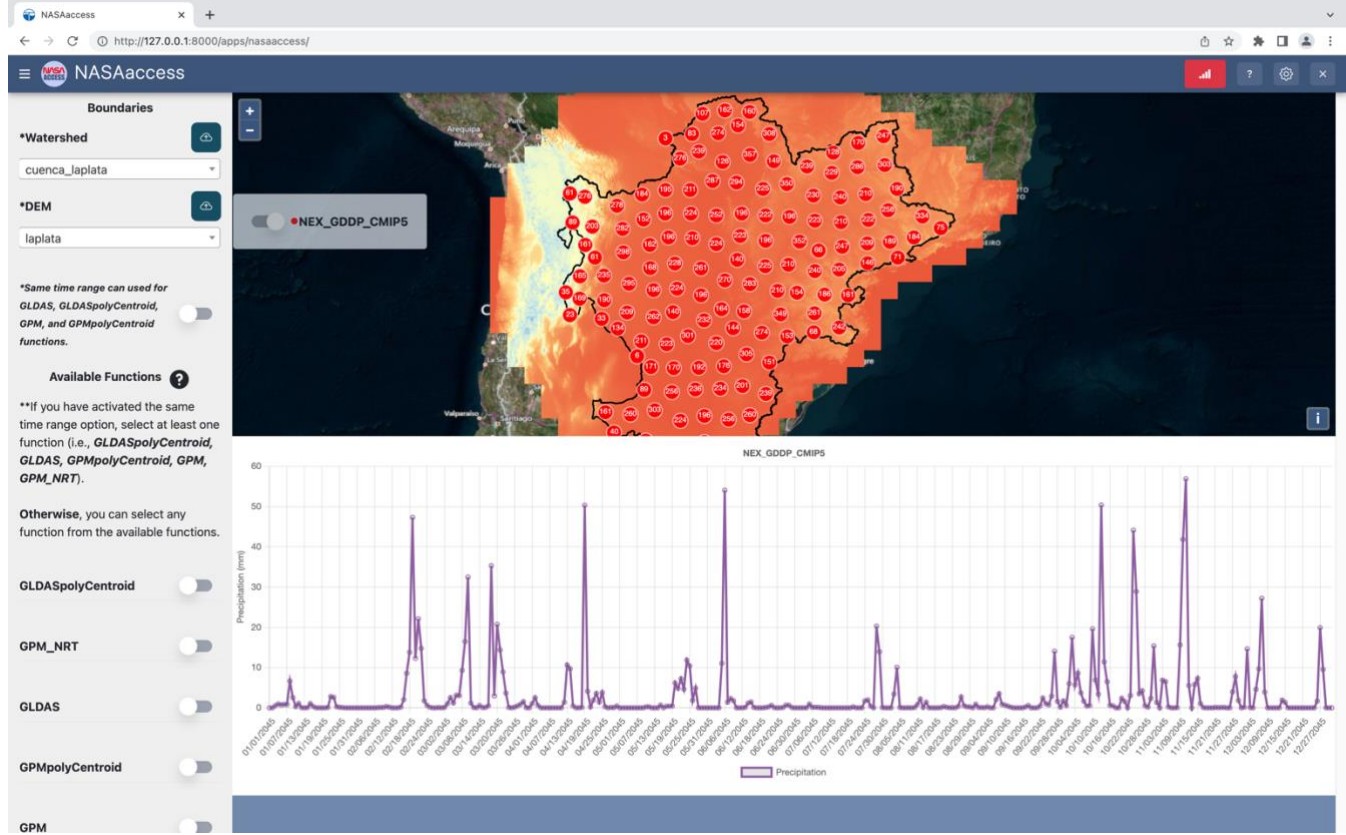

**Figure 11: The daily downscaled precipitation in millimeters projected by the GFDL-ESM2M GCM model across the rcp85**
**greenhouse gas emission at the La Plata Basin from 2045-January-01 to 2045-December-31 as obtained by the `NEX_GDDP_CMIP5`**
**function of the *NASAaccess* Tethys application. The `NEX_GDDP_CMIP5` generates downscaled daily precipitation and diurnal air**
**temperature data from the NASA CMIP5 downscaled climate change data products. The La Plata Basin depicted with digital**
**elevation model layer includes areas of southeastern Bolivia, southern and central Brazil, the entire country of Paraguay, most of**
**Uruguay, and northern Argentina. Map created and drafted using © Microsoft Bing Maps Platform API.**

975