# Peer review of "Technical note: NASAaccess – A tool for access, reformatting, and visualization of remotely sensed earth observation and climate data"

_EGUsphere, 2023_

## Author Comment (AC1)

Mohammed et al. response for egusphere-2023-328

The authors developed an open-source package to ease the access, reformat, and visualization of remote sensing earth observation data from NASA, to facilitate data dissemination, assist hydrological modeling, and support decision making. The package currently supports multiple NASA climate datasets, and the reformatted data could be seamlessly ingested into several mainstream hydrological models. The manuscript has clearly described the functionalities and methodology of the package, and given detailed instructions on installation requirements and steps, as well as how to use it with a case study. All the technical instructions are easy to follow. Also, the package has already been demonstrated in two published articles. Below are comments that suggest the authors to address.

Dear Referee,

We would like to extend our thanks to you for your valuable and constructive comments on our paper. In the revised manuscript, we have addressed all the comments raised. Specifically, we have revised Table 1 by adding more information to help the readers in comparing NASAaccess with existing tools. We have also added some information related to NASAaccess framework retrieval time. We think that our revised manuscript has been adjusted to add strength and support to our tool. Finally, we have addressed all the minor edits as requested to enhance reading the paper easily. Please find below our detailed response to the comments listed.

1.    What are the benefits of NASAaccess comparing with existing tools listed in Table 1? In Table 1, it would help if you could list more information of existing tools to illustrate the necessity and benefits of NASAaccess, such as open source or not, supported datasets, programming language, operation system, and other pros and cons related to the purpose of this work.

The main benefits for NASAaccess framework can be summarized as: 1) an-open source tool, 2) modular - which means the framework could be replicated, customized, and implemented anywhere, 3) seamless earth-observation remote sensing and climate data ingestion into other modeling frameworks – *NASAaccess gives ready formatted ascii data required to drive various hydrological models,* and 4) lowering the technical barrier for leveraging and visualizing a wide array of satellite-based earth observations. The above-mentioned points have been discussed in section 4 of the manuscript. In the revised manuscript we modified Table 1 by adding three columns (Visualization Capability; Data Retrieval Format; Source Code Availability) to illustrate the differences between NASAaccess and some of the current NASA GES DISC tools and services for accessing and visualizing earth observation remote sensing data as requested.

2.    For some abbreviations, please only give full name when one item is first time mentioned in the manuscript, for examples, NASA on Page 3 and 16, GeoGloWS on Page 17, GLDAS on Page 15, GES on Page 28, 29, and 30, DISC on Page 28, 29, and 30, CMIP on Page 18, 24, and 26 (twice), OSSI on Page 16, SWAT on Page 6, etc.

Addressed. Thanks!

**3. In Section 3.1, these three functions in NASAaccess are mentioned here for the first time. Please explain them here for readers to better understand or give a note to inform readers to find "further explanation in XXXX of the Appendix"?**

We added the following sentence to the revised manuscript in response to this comment as requested.

*'Further explanation of* `GPM_NRT`*,* `GPMpolyCentroid` *and* `GPMswat` *functions are listed in NASAaccess Documentation part of the Appendix.'*

**4. In Section 3.1, please give full name and explain the "data of IMERG". It would be better to give the website of IMERG data https://gpm.nasa.gov/data/imerg**

Web site address being changed as mentioned.

**5. Where can readers find the shapefile and DEM for the case study?**

The shapefile and DEM for the case study are packaged with NASAaccess R software version. This is mentioned in the scripts. In addition, we have added the following sentence to the revised manuscript as requested.

*'The readers can also find these data files at the NASAaccess OSF home page (https://osf.io/ctj2k/) 'extdata' section.'*

**6. Is there any limitation on how long the data could be retrieved? For years of data, how long it would take with different functions?**

Thanks for raising this point in the discussion. The limitation on record retrieval using NASAaccess relies on data availability at NASA servers. The test results paragraph shown below have been added to the revised manuscript.

There are multiple factors such as internet bandwidth (i.e., volume of information that can be sent over a connection in a measured amount of time), internet speed, and study site size that interact in figuring out the time duration of any NASAaccess function execution. To illustrate this further, here is an example for one month data record retrieval using the GPM_NRT function over the same study site shown above.

```
> system.time({ GPM_NRT(Dir = "./GPM_NRT_2/",
+              watershed = shape_path,
+              DEM = dem_path,
+              start = "2023-04-01",
+              end = "2023-04-30") })
**Results**
**user  system elapsed**
**30.023  21.869 130.313**
```

The results give "user", "system", and "elapsed" times. The "user" gives the CPU time spent by the current process (i.e., the current R session) in seconds and "system" gives the CPU time spent by the kernel (the operating system) on behalf of the current process. The "elapsed" is the wall clock time taken to execute the GPM_NRT function (i.e., 130.313 seconds). Upon checking the internet speed utilized on (Intel(R) Core(TM) i9-9880H CPU @ 2.30GHz) machine, it reveals:

==== SUMMARY ====

Upload capacity: 17.543 Mbps
Download capacity: 107.578 Mbps
Upload flows: 12
Download flows: 12
Responsiveness: Medium (714 RPM)

---

## Author Comment (AC2)

Mohammed et al. response for egusphere-2023-328

This paper presents novel open-source software packages and web-based environmental modeling applications for Earth observation data accessing, reformation, and presenting quantitative data products. This software is very useful for the scientists and stakeholders to further provide support for environmental modeling . This software can help to lower the technical barriers and leverage the distributed computing resources for environmental modeling. The user manuals are described very well and easily to follow. And the paper is well written.

Dear Referee,
We would like to extend our thanks to you for your valuable and constructive feedback on our paper. In the revised manuscript, we have addressed all the comments raised. Specifically, we have revised Table 1 by adding more information to help the readers in comparing NASAaccess with existing tools. We have also added some information related to NASAaccess framework retrieval time. We think that our revised manuscript has been adjusted to add strength and support to our tool. Finally, we have addressed all the minor edits as requested to enhance reading the paper easily. Please find below our detailed response to the comments listed.

However, I have some suggestions/questions:

1. There are some similar functionality between the NASAaccess and other open-sources mentioned in Table 1. What is the gap between NASAaccess and other software in the Table? What benefit can we obtain by using NASAaccess compared to other software?

The main benefits for NASAaccess framework can be summarized as: 1) an-open source tool, 2) modular - which means the framework could be replicated, customized, and implemented anywhere, 3) seamless earth-observation remote sensing and climate data ingestion into other modeling frameworks – NASAaccess gives ready formatted ascii data required to drive various hydrological models, and 4) lowering the technical barrier for leveraging and visualizing a wide array of satellite-based earth observations. The above-mentioned points have been discussed in section 4 of the manuscript. In the revised manuscript we modified Table 1 by adding three columns (Visualization Capability; Data Retrieval Format; Source Code Availability) to illustrate the differences between NASAaccess and some of the current NASA GES DISC tools and services for accessing and visualizing earth observation remote sensing data as requested.

2. Line 130 - 135, once the data were generated and downloaded, what kind of data format it would be? And could you show some examples?This could be important information for the end-user to further use the processed data.

Once the data was generated and downloaded, it would be in gridded ascii format suitable for ingestion by various hydrological models. We have presented some examples in section 3 showing the gridded ascii format layout. Yes, we do agree that this is important information for end-users since most existing tools do not give ascii output formatted to be ingested with other hydrological models.

3. **Line 175-176. The authors mentioned the .netrc file for storing the credentials information. What is the".urs_cookies" file for?**

This ".urs_cookies" file will be used to persist sessions across individual cURL calls, making it more efficient. The revised manuscript has been updated with this comment response.

4. **Line 175-177 is the same contents as 185-190**

Yes. The repeated content has been removed in the revised manuscript.

5. **I am not sure if I understand correctly. The raw data is saved in NASA' server with PostgreSQL database? The users can download the data which is not saved in the PostgresQL as mentioned in Line 114 . Then why the database is required to be downloaded by the database server as stated in Line 200 - 205. Indeed, it would be great if the authors can add a data flow and data information , i.e. where is the raw data, what are formats of the data, how the data is processed, where is the processed data stored,what is the processed data format etc.**

The Tethys framework comes with PostgresQL database to store data as a standard configuration. The NASAaccess web-based application does not create tables and store them in the PostgresQL database associated with the Tethys application standard practice. Rather, we have designed the NASAaccess web-based application to let the user download the data when it is ready immediately rather than writing the data in tables. This would save time in executing jobs using the NASAaccess web-based application. Line 113-114 in the revised manuscript says that the NASAaccess web-based application fetches and retrieves data without saving it in the PostgresQL associated with the Tethys framework. It is also important to point out that the Tethys Platform allows the user to choose if the app that one is going to develop has a PostgreSQL database integrated with it. In the case of the NASAaccess web application, no PostgreSQL database is integrated with it.

6. **The tool seems to be supported by R and Python. But the examples in the manuscript are only given to R. I would like to suggest authors to add examples that use Python. Or you can point to the document link where there is any example for Python**

Yes, NASAaccess conda package supports R and Python because of the nature of conda which is language-agnostic binary package manager. All the examples given in section 3 can be replicated using Python by calling the Rscript executable in any conda environment that has the conda package NASAaccess installed. Lines 488 – 492 in the revised manuscript show an example that uses Python with the NASAaccess tool.